# FoleyGenEx: Unified Video-to-Audio Generation with Multi-Modal Control, Temporal Alignment, and Semantic Precision

## Abstract

We introduce FoleyGenEx, a unified framework for video-to-audio (VTA) generation that integrates multi-modal control, frame-level temporal alignment, and fine-grained semantic expressivity, enabling synchronized, versatile, and expressive audio synthesis across diverse tasks. Existing VTA methods either offer multi-modal control with weak temporal alignment or achieve strong alignment while lacking reference audio conditioning and semantic precision. FoleyGenEx bridges this gap through three key innovations: a conditional injection mechanism enabling audio-controlled VTA and Foley extension, a multi-modal dynamic masking strategy preserving synchronization during multi-modal training, and an adverb-based data augmentation algorithm leveraging signal processing and large language models to enrich audio representations and textual supervision with nuanced semantic cues. Experiments on AudioCaps, VGGSound, and Greatest Hits show that FoleyGenEx delivers competitive performance in controllable VTA generation, achieving strong temporal fidelity, versatile multi-modal control, and fine-grained semantic precision compared to existing methods. Demo samples are available at `https://foleygenex.github.io/FoleyGenEx`.

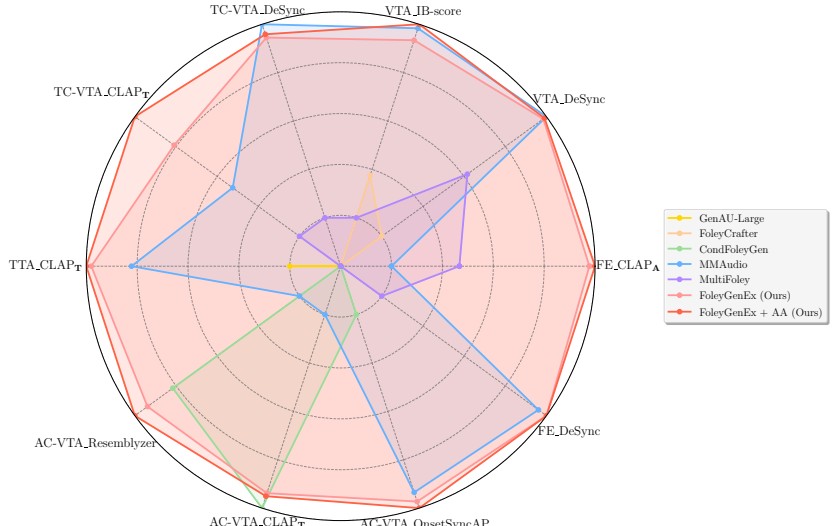

Figure 1: FoleyGenEx supports a range of multi-modal controlled audio generation tasks, including Text-to-Audio (TTA), basic Video-to-Audio (VTA), Text-Controlled VTA (TC-VTA), Audio-Controlled VTA (AC-VTA), and Foley extension (FE). It unifies these tasks while achieving strong synchronization, versatile control, and expressive audio generation.

## 1 Introduction

Text-to-video (T2V) generation has made remarkable progress with models such as OpenAI's Sora (OpenAI, 2024) and others (Yang et al., 2024; Kong et al.; Kuaishou, 2024; Wan et al., 2025), producing visually compelling content. However, most T2V outputs remain silent, severely limiting

user immersion and realism. Adding synchronized audio would dramatically enhance the viewing experience, yet manual dubbing and alignment are prohibitively time-consuming. This challenge has driven growing interest in generative models for video-to-audio (VTA) synthesis, which aim to automatically produce audio tracks conditioned on video content.

Despite recent advances, existing VTA methods face persistent challenges in three key areas: temporal synchronization, multi-modal control, and fine-grained semantic precision. MultiFoley (Chen et al., 2025), for instance, integrates video, text, and reference audio to enable style transfer and Foley extension (FE), but relies on a simplistic strategy of upsampling video features for alignment, leading to degraded synchronization. In contrast, MMAudio (Cheng et al., 2024) employs a multi-modal diffusion transformer (MMDiT) (Esser et al., 2024) with Synchformer (Iashin et al., 2024) to achieve strong frame-level synchronization, yet lacks reference audio conditioning and therefore struggles with tasks such as audio-controlled VTA (AC-VTA) and FE, where the goal is to generate synchronized audio matching the timbre, prosody, and audio event of a reference track. Moreover, both approaches fail to provide fine-grained semantic control when textual descriptions specify subtle variations in manner or intensity—for example, distinguishing between "fast knocking" and "slow knocking," or "loud knocking" and "soft knocking"—largely because standard datasets (Chen et al., 2020; Kim et al., 2019; Mei et al., 2024) contain few adverbial cues, offering limited supervision for such nuanced semantics.

These limitations create a clear trade-off: MultiFoley supports diverse control inputs but sacrifices temporal precision, whereas MMAudio delivers strong synchronization but lacks versatility in conditioning and semantic expressivity. As illustrated in Figure 1, neither approach achieves all three objectives simultaneously, leaving a gap for methods that can unify temporal alignment, multi-modal control, and semantic precision within a single framework.

To close this gap, we introduce FoleyGenEx, a novel VTA framework built upon the MMDiT architecture. FoleyGenEx preserves the strong synchronization capabilities of MMAudio while extending its functionality with a conditional injection mechanism that enables reference audio conditioning for AC-VTA and FE. A multi-modal dynamic masking strategy further ensures alignment between training and inference, mitigating the synchronization degradation observed in MultiFoley. Beyond synchronization and control, FoleyGenEx incorporates an adverb-based data augmentation algorithm that leverages both signal processing techniques and large language models (LLMs) to enrich training data with adverbial cues, enabling fine-grained semantic control over the generated audio.

Extensive experiments on AudioCaps (Kim et al., 2019), VGGSound Chen et al. (2020), and Greatest Hits Owens et al. (2016) demonstrate that FoleyGenEx not only inherits the strong synchronization performance of MMAudio but also achieves the versatile multi-modal control of MultiFoley, while uniquely providing fine-grained semantic precision through adverb-based augmentation. As shown in Figure 1, FoleyGenEx and its adverb-augmented variant achieve competitive performance compared to existing methods across synchronization, control, and semantic metrics, and excel in several key aspects, representing significant progress for controllable VTA generation.

## 2 RELATED WORK

**Flow matching.** Flow matching (Lipman et al., 2022) models the continuous transformation of data distributions using an ordinary differential equation (ODE):

$$\frac{dx_\tau}{d\tau} = v(\tau, x_\tau),$$

where $x_\tau$ represents the state at continuous time $\tau$ (ranging from 0 to 1), and $v(\tau, x_\tau)$ is a learned flow field guiding $x_\tau$ from an initial distribution $p_0(x)$ (e.g., Gaussian noise) to a target distribution $p_1(x)$ (the real data distribution). To solve this ODE numerically, Euler's method is used for discretization, resulting in the iterative update formula:

$$x_t = t \cdot x_1 + (1 - t) \cdot x_0,$$

where $t$ is a discrete time step (a scalar controlling interpolation), $x_0$ is a sample from the initial noise distribution $p_0(x)$, and $x_1$ is a sample from the target data distribution $p_1(x)$.

**Multi-modal Diffusion Transformer.** MMDiT Esser et al. (2024) builds upon the Diffusion Transformer (DiT) architecture, emphasizing collaborative modeling and information interaction across

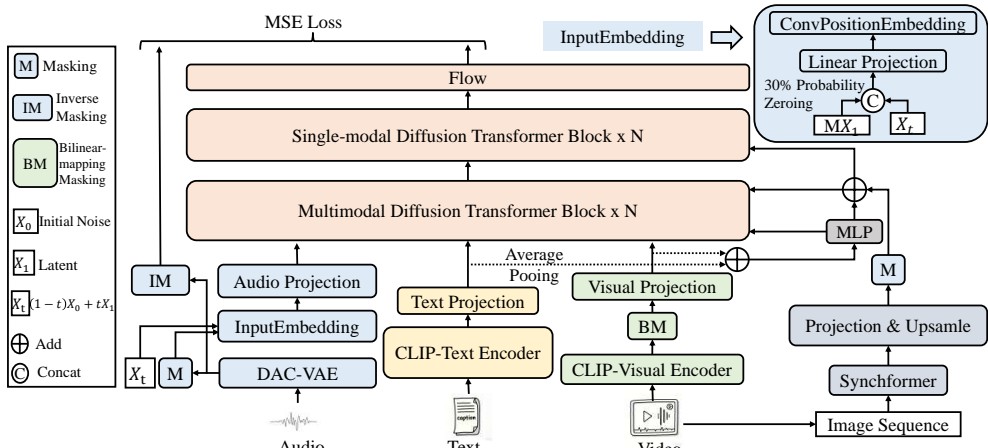

Figure 2: FoleyGenEx training framework.

multiple modalities. This is crucial for cross-modal generative tasks like VTA, which require coordinated multi-source conditioning. MMDiT uses separate weight parameters to construct independent feature streams for each modality. Within attention layers, it concatenates multi-modal sequences to enable bidirectional cross-modal information flow, preserving modality-specific characteristics while ensuring precise alignment. To further enhance cross-modal collaboration, the adaLN mechanism Perez et al. (2018) is integrated to inject conditions.

**Multi-Modal Controlled Audio Generation.** Text-to-audio (TTA) (Liu et al., 2024a; Deepanway et al., 2023; Majumder et al., 2024; Huang et al., 2023b;a; Haji-Ali et al., 2024) offers control over background and environmental sounds but often struggles with video semantics or accurate synchronization. Conversely, VTA (Viertola et al., 2025; Liu et al., 2024b; Wang et al., 2024b; Zhang et al., 2024; Wang et al., 2024a; Cheng et al., 2024) excels at generating audio aligned with video content and events, essential for applications like silent film dubbing and damaged audio track restoration. Recent efforts focus on multi-modal controlled audio generation to increase VTA's flexibility. For example, CondFoleyGen (Du et al., 2023) uses target video, conditional video, and conditional audio to generate synchronized audio, demonstrating strong style transfer and beat-syncing on the Greatest Hits dataset. Sketch2Sound (García et al., 2025) uses loudness, spectral centroid, and pitch probabilities from reference audio to control the generated audio's prosody. MultiFoley integrates video, text, and audio, force-aligning video features with audio latent vectors for multi-modal control, including style transfer and FE, achieving comprehensive control.

## 3 FOLEYGENEX

### 3.1 OVERVIEW

Our FoleyGenEx training framework, illustrated in Figure 2, leverages MMDiT for multi-modal information fusion and a single-modal DiT for audio modeling using iterative flow-matching. A key innovation is a conditional injection mechanism in the audio branch, combined with a novel multi-modal dynamic masking strategy applied across modalities.

**Audio Modality** Audio latents are extracted using DAC-VAE Kumar et al. (2023). To enable conditional information injection and promote robust zero-shot generation, a random masking strategy is employed, masking 70-100% of the audio latent. This aligns the training process ("context + mask") with the inference process ("reference audio + generated audio"). During iterative flow-matching, an InputEmbedding layer processes both the masked latent and the intermediate state $X_t$ at each time step to inject conditional information. Specifically, the masked latent is zeroed out with a 30% probability, and then this (potentially zeroed) masked latent is concatenated with $X_t$ along the channel dimension before being projected through a linear layer. This projected output is then processed by a convolutional positional embedding module (ConvPositionEmbedding) Le et al. (2023) to add sequence positional information, and the resulting fused features are projected to the input dimension of MMDiT via an audio projection layer. At inference, the conditional latent is injected via the InputEmbedding layer and concatenated before the initial noise, effectively making the entire

initial noise segment functionally equivalent to the masked portion used during training. To ensure the generated audio's duration synchronizes with the video length within the MMDiT architecture, a 'fake' reference video is concatenated before the target video. To mitigate the influence of potentially misaligned semantic and synchronization features due to the reference audio's misalignment with this 'fake' reference video, mask processing is also applied to the video branch, as detailed in a later section.

**Text Modality** Semantic features are extracted from text using the CLIP (Radford et al., 2021) text encoder. Since text and audio are not temporally aligned, masking is not applied to text features.

**Video Modality** The video's semantic features are extracted using the CLIP visual encoder. Synchronization features are extracted using Synchformer, processing the video as a sequence of images. To mitigate the influence of potentially misaligned semantic and synchronization features from the 'fake' reference video, the video semantic features undergo bilinear mapping-based masking, corresponding to the mask applied to the audio latents. We also explored an alternative approach: upsampling the video semantic features before applying the same masking procedure used for the audio latents. However, this method resulted in suboptimal performance, likely due to the padding introduced during upsampling, which dilutes or alters the original video semantic information. The synchronization features are projected and upsampled to align with the audio's temporal dimension before undergoing the same masking process as the audio latents. This strategy ensures that even when reference videos are unsynchronized, they do not interfere with the injection of reference audio information or the alignment between the generated audio and the target video.

**Conditioning** The video and text semantic features are projected and concatenated with the audio latents before being jointly input into the MMDiT for multi-modal information fusion. In parallel, these semantic features are average-pooled and projected using a multi-layer perceptron (MLP) to create a global conditioning vector. This global condition is then combined with the video synchronization features to construct a frame-aligned condition. Both the global and frame-level conditions are injected into the MMDiT blocks via the adaLN mechanism. And the frame-aligned condition is also injected into the single-modal DiT.

Finally, loss computation is restricted to the masked segments, ensuring that learning focuses on the critically aligned content.

### 3.1.1 DISTINCTION FROM MMAUDIO ARCHITECTURE

To clarify the boundary between FoleyGenEx and the base MMAudio framework, we explicitly detail the key innovations and modifications:

- Conditional Injection Mechanism: FoleyGenEx introduces a novel conditional injection mechanism within the audio branch, a feature entirely absent in MMAudio. This mechanism integrates conditional latent and intermediate states through channel-wise concatenation, leverages the InputEmbedding module for reference audio conditioning, and incorporates summation during inference.

- Multimodal Masking: Unlike MMAudio, which lacks any masking design, FoleyGenEx implements a comprehensive multimodal masking strategy across its audio, video semantic, and synchronization branches. This ensures train-inference consistency by applying targeted masking: 70-100% random masking for audio and bilinear mapping/synchronized masking for video branches.

- Loss Adaptations: FoleyGenEx replaces MMAudio's standard Mean Squared Error (MSE) loss with a masked MSE loss, which specifically targets and focuses on aligned segments.

### 3.2 MULTI-MODAL-CONTROLLED AUDIO GENERATION

The flexible adaptation of our FoleyGenEx architecture to various input modality features enables five distinct types of multi-modal controlled audio generation, as illustrated in Figure 3. These include text-to-audio (TTA), video-to-audio (VTA), text-controlled video-to-audio (TC-VTA), audio-controlled video-to-audio (AC-VTA), and Foley extension (FE). Furthermore, leveraging the latent inversion property (Le Lan et al., 2024) of DiT, we achieve a sixth task: fine-grained audio editing by controlling the generation of segments in specific time regions.

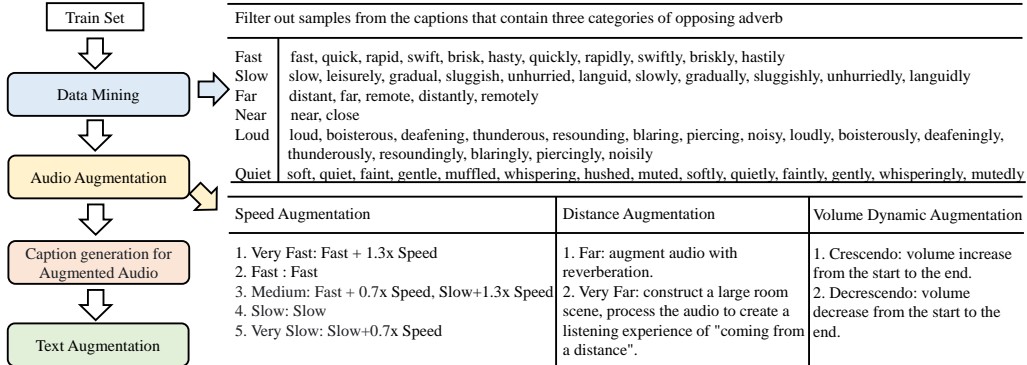

Figure 3: Multi-modal controlled audio generation tasks. $F_T$: text semantic features; $F_V$: video semantic features; $F_S$: video synchronization features; $F_{RS}$: reference video synchronization features; $F_A$: reference audio latents. $F_{V0}$, $F_{A0}$, and $F_{T0}$ represent all-zero features for video, audio, and text, respectively. 'Ref' denotes 'reference'.

| Train Set | Filter out samples from the captions that contain three categories of opposing adverb | | |
|---|---|---|---|
| Data Mining | Fast | fast, quick, rapid, swift, brisk, hasty, quickly, rapidly, swiftly, briskly, hastily | |
| | Slow | slow, leisurely, gradual, sluggish, unhurried, languid, slowly, gradually, sluggishly, unhurriedly, languidly | |
| | Far | distant, far, remote, distantly, remotely | |
| | Near | near, close | |
| Audio Augmentation | Loud | loud, boisterous, deafening, thunderous, resounding, blaring, piercing, noisy, loudly, boisterously, deafeningly, thunderously, resoundingly, blaringly, piercingly, noisily | |
| | Quiet | soft, quiet, faint, gentle, muffled, whispering, hushed, muted, softly, quietly, faintly, gently, whisperingly, mutedly | |
| Caption generation for Augmented Audio | Speed Augmentation | Distance Augmentation | Volume Dynamic Augmentation |
| | 1. Very Fast: Fast + 1.3x Speed 2. Fast : Fast 3. Medium: Fast + 0.7x Speed, Slow+1.3x Speed 4. Slow: Slow 5. Very Slow: Slow+0.7x Speed | 1. Far: augment audio with reverberation. 2. Very Far: construct a large room scene, process the audio to create a listening experience of "coming from a distance". | 1. Crescendo: volume increase from the start to the end. 2. Decrescendo: volume decrease from the start to the end. |
| Text Augmentation | | | |

Figure 4: Adverb-based data augmentation algorithm.

**TTA** Only the semantic feature of the text is provided; all other features are set to zero. This generates audio aligned with the text's semantic meaning.

**VTA** The semantic and synchronization features of the target video are provided. The text semantic feature is optional, but if provided, it should be relevant to the video content. This setup generates audio that is temporally synchronized with the target video and matches its semantics.

**TC-VTA** The video semantic feature is set to zero. The text semantic feature and video synchronization feature guide audio generation. Notably, the text content does not need to be relevant to the video. For example, a video of a cat meowing with the text "Lion roaring" can generate a lion's roar synchronized with the cat's meows.

**AC-VTA** Both video and text semantic features are set to zero. The latent derived from the reference audio is not only concatenated with the initial noise along the channel dimension but also summed with it. Additionally, a 'fake' reference video is constructed by cropping or duplicating the target video to match the duration of the reference audio. The synchronization features of this fake reference video are then placed at the position corresponding to the reference audio latent—specifically, concatenated before the synchronization features of the target video. This task utilizes the timbre, prosody, and audio events of the reference audio as semantic cues, which are combined with the target video's synchronization features to guide audio generation. For instance, it enables generating metal-knocking audio that synchronizes with vegetable-chopping actions in the target video, with a metal-knocking audio clip serving as the reference.

**FE** The semantic and synchronization features of the target video are provided. Additionally, the latent extracted from a segment of the video's existing audio is summed with the initial noise for conditional guidance. This generates audio that matches the video and continues the style of the provided audio segment.

**Editing** Building upon TC-VTA, AC-VTA, and FE, we leverage DiT's latent inversion to enable fine-grained audio editing. This allows users to regenerate audio segments within specific time regions and seamlessly concatenate them with unedited portions in the latent space, avoiding clipping artifacts from waveform-level concatenation. This local editing feature refines initial audio generation results, as demonstrated by examples on our website.

Table 1: Performance on the AudioCaps test set. The results of all baseline are from Cheng et al. (2024). Bold font indicates the optimal performance under each metric, while underlined text marks the second-best performance.

| Method | $FD_{VGG} \downarrow$ | $IS \uparrow$ | $CLAP_T \uparrow$ |
|---|---|---|---|
| GenAU-Large (Haji-Ali et al., 2024) | **1.21** | 11.75 | 0.285 |
| MMAudio (Cheng et al., 2024) | 4.03 | **12.08** | 0.348 |
| FoleyGenEx (Ours) | 2.61 | 11.80 | 0.364 |
| FoleyGenEx + AA (Ours) | 2.60 | 11.89 | **0.366** |

### 3.3 ADVERB-BASED DATA AUGMENTATION

While our framework improvements expand the functionality of multi-modal-controlled audio generation, we also recognize the crucial role of data in enabling fine-grained control. From the tasks introduced in Section 3.2, we observe that text serves as a key semantic control condition. However, existing models tend to prioritize nouns and actions in text, neglecting control over adverbs (e.g., those specifying speed, distance, and volume), which significantly impact the generated audio. We attribute this shortcoming to the scarcity of adverb-related training data, and the prohibitively high cost of manual annotation. To address this limitation and achieve refined semantic control of adverbs within texts for generated audio, we propose an adverb-based data augmentation algorithm (Figure 4), which comprises four key steps:

**Data Mining** We focus on three categories of opposing adverbs: speed ('Fast/Slow'), distance ('Far/Near'), and volume ('Loud/Quiet'). Samples containing adverbs from these categories (or their corresponding adjective forms) are filtered to create the base dataset.

**Audio Augmentation** We apply targeted audio augmentations based on the initial adverb.

- Speed Augmentation: For samples with speed adverbs, we construct a five-level scale by adjusting audio speed: (1) Very Fast: Original 'Fast' samples, speed increased by 1.3x. (2) Fast: Original 'Fast' samples. (3) Medium: Original 'Fast' samples slowed by 0.7x; original 'Slow' samples speed up by 1.3x. (4) Slow: Original 'Slow' samples. (5) Very Slow: Original 'Slow' samples, speed reduced by 0.7x.

- Distance Augmentation: For 'Near' or 'Loud' samples (typically corresponding to higher volume audio), we apply distance augmentation.

- Volume Dynamic Augmentation: For 'Loud' samples, we adjust the audio volume trend to achieve dynamic volume augmentation, while maintaining the original maximum volume.

**Caption Generation for Augmented Audio** We use LLMs with specifically designed prompts (available on our website) to generate new captions for the augmented audio. These prompts leverage both the original caption and the specific audio augmentation method applied.

**Text Augmentation** To enhance generalization, we use LLMs to extract keywords from the augmented captions and generate paraphrased versions. This approach acknowledges that a single audio clip can be described by multiple text descriptions with varying phrasings but the same meaning.

We manually verified the correspondence between augmented audio and generated captions for a randomly sampled set of 300 entries. The sampling was stratified: 20 entries per speed level (across 5 levels), and 50 entries per operation type for both distance and volume augmentations (2 types each). This verification process achieved an accuracy rate exceeding 97%.

## 4 EXPERIMENTAL SETTINGS

### 4.1 TRAIN DATASET

Our model was trained on a combined dataset of VTA and TTA data. The base training data configuration, consistent with MMAudio, includes approximately 500 hours of VTA data from VGGSound, along with TTA data comprising roughly 128 hours from AudioCaps and 7,600 hours from WavCaps (Mei et al., 2024). Additionally, we incorporated adverb-augmented data (88,370 samples) generated through a multi-step process (see Appendix B for details) in specific training runs. The identifier 'AA' indicates training runs that include this adverb-augmented data in the TTA subset.

Table 2: Performance on the VGGSound test set.

| Model | FD$_{VGG}$ ↓ | FD$_{PASST}$ ↓ | IS ↑ | IB-score ↑ | DeSync ↓ |
|---|---|---|---|---|---|
| VTA-LDM (Xu et al., 2024) | 2.04 | 173.36 | 10.14 | 24.72 | 1.263 |
| FoleyCrafter (Zhang et al., 2024) | 2.51 | 140.09 | 15.68 | 25.68 | 1.225 |
| MMAudio (Cheng et al., 2024) | 0.97 | 60.60 | 17.40 | **33.22** | **0.442** |
| FoleyGenEx (Ours) | 0.74 | 46.67 | 18.37 | 32.52 | 0.467 |
| FoleyGenEx + AA (Ours) | **0.73** | **45.69** | **18.49** | 33.17 | 0.453 |
| MultiFoley * (Chen et al., 2025) | 2.92 | — | — | 28.00 | 0.800 |
| VTA-LDM * | 2.02 | 171.04 | 11.32 | 28.34 | 1.275 |
| FoleyCrafter * | 2.74 | 140.12 | 16.15 | 30.20 | 1.240 |
| MMAudio * | 1.13 | 61.59 | 17.59 | 37.85 | **0.393** |
| FoleyGenEx (Ours) * | 0.87 | 47.81 | 18.64 | 37.23 | 0.409 |
| FoleyGenEx + AA (Ours) * | **0.86** | **47.20** | **19.56** | **38.06** | 0.403 |

## 4.2 TRAINING AND EVALUATION SETUP

Our model architecture and training procedure were implemented based on the MMAudio framework[1]. We adopted the learning rate, scheduler, and MMAudio-L-44.1kHz hyperparameter configuration from the original MMAudio settings and used a batch size of 256. Following the MMAudio protocol, base training was conducted for 300,000 steps, requiring three days on a machine equipped with 8 A100 GPUs. When training with the additional 88,370 adverb-augmented TTA data samples, we extended the training to 330,000 steps to maintain a comparable data traversal, accounting for the approximate 10% increase in dataset size. To facilitate classifier-free guidance (CFG) Ho & Salimans (2022) during inference, we randomly dropped either video or text features with a 10% probability, replacing them with all-zero feature vectors.

We evaluated performance across four dimensions using metrics from the av-benchmark toolkit[2]: distribution matching, audio quality, semantic alignment, and temporal alignment. Distribution matching was assessed by calculating Fréchet Distance (FD) between generated and real audio features, using PaSST (Koutini et al., 2021) (FD$_{PaSST}$) and VGGish (Hershey et al., 2017) (FD$_{VGG}$) as embedding models. Audio quality was evaluated via Inception Score (IS) (Salimans et al., 2016) with PANNs (Kong et al., 2020). Semantic alignment was evaluated differently for TTA and VTA tasks: for TTA, average cosine similarity (CLAP$_T$) was calculated using CLAP (Wu et al., 2023)-extracted text and generated audio features; for VTA, average cosine similarity (IB-score) computed via ImageBind (Girdhar et al., 2023)-extracted visual and generated audio features. Temporal alignment was measured by audio-video misalignment in seconds predicted by Synchformer (DeSync).

## 5 RESULTS AND DISCUSSIONS

### 5.1 TTA & VTA

For TTA task evaluation, we used the AudioCaps test set (964 samples), and for VTA task evaluation, we used the VGGSound test set (15,220 samples), with metadata provided by MMAudio. To ensure a fair comparison with MultiFoley, whose test set excludes VGGSound samples with an IB-score below 0.3, we filtered the VGGSound test set accordingly, resulting in a test set of 8702 samples, consistent with MultiFoley. In Table 2, results marked with '*' are evaluated on this filtered 8702-sample test set; otherwise, the results are evaluated on the complete VGGSound test set. We extracted the corresponding 8702 samples from the publicly available MMAudio inference results and evaluated their metrics. We also performed inference evaluations using FoleyCrafter[3] and VTA-LDM[4]. For all reproduced models and our own model, we generated an 8-second audio clip for each video during inference. Specifically, our model's inference settings included diffusion sampling with a guidance scale of 4.5 and 25 inference steps. The evaluation results for the TTA and VTA tasks are detailed in Tables 1 and 2, respectively. For TTA, our method outperforms MMAudio in both distribution matching and semantic relevance between audio and text, while maintaining competitive

---

[1] https://github.com/hkchengrex/MMAudio

[2] https://github.com/hkchengrex/av-benchmark

[3] https://github.com/open-mmlab/FoleyCrafter

[4] https://github.com/ariesssxu/vta-ldm

Table 3: Comparison of text-controlled video-to audio performance.

| Method | DeSync ↓ | CLAP$_T$ ↑ |
|---|---|---|
| MultiFoley (Chen et al., 2025) | 0.169 | 31.39 |
| MMAudio (Cheng et al., 2024) | **0.053** | 32.53 |
| FoleyGenEx (Ours) | 0.061 | 33.53 |
| FoleyGenEx + AA (Ours) | 0.059 | **34.20** |

Table 4: Comparison of audio-controlled video-to audio performance.

| Method | OnsetSyncAP(%) ↑ | FD$_{VGG}$ ↓ | Resemblyzer ↑ | CLAP$_A$ ↑ |
|---|---|---|---|---|
| CondFoleyGen (Du et al., 2023) | 60.00 | 0.65 | 0.8999 | **0.7385** |
| MMAudio-S1 | 65.53 | 4.77 | 0.8185 | 0.5063 |
| MMAudio-S2 | 68.72 | 3.05 | 0.8360 | 0.5132 |
| MMAudio-S3 | 68.92 | 2.16 | 0.8568 | 0.5195 |
| FoleyGenEx (Ours) | 69.38 | **0.54** | 0.9085 | 0.7216 |
| FoleyGenEx + AA (Ours) | **69.71** | **0.54** | **0.9128** | 0.7250 |

audio quality. Similarly, for VTA, we observe improvements over MMAudio in distribution matching and audio quality, alongside competitive audio-video relevance and temporal alignment. These results suggest that the conditional injection mechanism and multi-modal dynamic mask training effectively expand the model's functionality without compromising performance on fundamental TTA and VTA tasks. Moreover, compared to MultiFoley, our method demonstrates significant advantages across all compared dimensions.

## 5.2 TC-VTA

In the TC-VTA task, the input text is semantically irrelevant to the video. Adapting both MMAudio and our model from the MMDiT framework, we set the video semantic feature input to all-zero feature. We utilized 13 TC-VTA examples from the MultiFoley website[5], using the video and text as conditions to generate audio with both models. We computed CLAP$_T$ and DeSync to compare semantic relevance and temporal alignment. As shown in Table 3, both MMAudio and our model outperform MultiFoley in temporal alignment and audio-text semantic relevance (The generated results of both MultiFoley and our FoleyGenEx have been made available on our website for result verification). This superiority demonstrates the effectiveness of the MMDiT framework, primarily due to its design: unlike MultiFoley, which directly upsamples video semantic features to achieve temporal alignment, MMDiT employs a more refined approach by processing the semantic and synchronization features of video separately.

## 5.3 AC-VTA

For the AC-VTA task, we followed the evaluation setup established by CondFoleyGen, which uses a test set derived from the Greatest Hits dataset. This involves generating 582 audio outputs by applying style transfer to 194 two-second Greatest Hits samples, using three different two-second videos as style references for each. Given that MultiFoley focuses on FE and does not provide AC-VTA evaluation, and due to the absence of publicly available code or models for MultiFoley preventing a fair comparison under identical settings, our AC-VTA comparisons were limited to CondFoleyGen, MMAudio, and our FoleyGenEx. For comparison, we utilized CondFoleyGen's open-source code[6] and pre-trained model (trained on non-overlapping Greatest Hits data), along with the pre-trained MMAudio-L-44.1kHz model. To address the lack of reference audio conditioning in the original MMAudio framework, we concatenated the latent of the reference audio with the initial noise and the intermediate state $X_t$, incorporating it as a conditional guide to enable explicit reference audio conditioning. As MMAudio generates audio based on input video length, we explored three setups to compensate for the missing video corresponding to the reference audio: (1) a two-second blank video concatenated before the target video (MMAudio-S1), (2) the first two seconds of the target video duplicated and prepended (MMAudio-S2), and (3) the reference video concatenated before the target video (MMAudio-S3). We evaluated performance across temporal alignment (Average Precision of Onset Synchronization, Onset Sync AP), distribution matching (FD$_{VGG}$), style

---

[5]https://ificl.github.io/MultiFoley/
[6]https://github.com/XYPB/CondFoleyGen/tree/main

Table 5: Comparison of Foley extension performance. $C_A$:$CLAP_A$.

| Sample Size | Conditions | | | | MultiFoley | | MMAudio | | FoleyGenEx | | FoleyGenEx + AA | |
|---|---|---|---|---|---|---|---|---|---|---|---|---|
| | $V_t$ | Text | $A_r$ | $V_r$ | $C_A$ ↑ | DeSync ↓ | $C_A$ ↑ | DeSync ↓ | $C_A$ ↑ | DeSync ↓ | $C_A$ ↑ | DeSync ↓ |
| **1000** | ✓ | ✓ | | | 55.4 | 0.79 | 56.0 | 0.40 | 59.7 | 0.39 | **60.5** | **0.38** |
| | ✓ | | ✓ | | 59.6 | 0.78 | 57.0 | 0.46 | 61.8 | **0.40** | **62.3** | 0.40 |
| | ✓ | | ✓ | ✓ | 59.8 | 0.77 | 59.0 | 0.39 | 69.3 | 0.37 | **69.7** | 0.36 |
| | ✓ | ✓ | ✓ | ✓ | 64.3 | 0.77 | 60.7 | 0.38 | 71.2 | 0.36 | **71.5** | 0.36 |
| **ALL** | ✓ | ✓ | | | — | — | 50.7 | 0.41 | 51.6 | 0.39 | **51.8** | 0.39 |
| | ✓ | | ✓ | | — | — | 52.8 | 0.47 | 62.7 | 0.40 | **62.9** | 0.39 |
| | ✓ | | ✓ | ✓ | — | — | 53.6 | 0.39 | 63.5 | 0.38 | **63.7** | 0.37 |
| | ✓ | ✓ | ✓ | ✓ | — | — | 57.4 | 0.38 | 65.8 | **0.36** | **65.9** | 0.36 |

Table 6: Performance of adverb-augmented data added to the training set.

| Setting | $FD_{VGG}$ ↓ | $KL_{PANNS}$ ↓ | $KL_{PASST}$ ↓ | $CLAP_T$ ↑ |
|---|---|---|---|---|
| MMAudio | 7.67 | 1.59 | 1.67 | 0.323 |
| FoleyGenEx (Ours) | 3.07 | 1.45 | 1.49 | 0.354 |
| MMAudio+AA | 4.03 | 1.47 | 1.45 | 0.361 |
| FoleyGenEx (Ours)+AA | **2.96** | **1.41** | **1.43** | **0.365** |

similarity—specifically timbre and prosody—(assessed using a Resemblyzer-based approach[7]), and semantic consistency (CLAP-based cosine similarity, $CLAP_A$), which measures the cosine similarity between CLAP embeddings of the reference and generated audio. As demonstrated in Table 4, our model exhibits strong performance in temporal alignment, distribution matching, and style similarity. While its $CLAP_A$ score is slightly lower than CondFoleyGen's, this is a noteworthy result, considering CondFoleyGen benefited from training on in-domain data. Among the MMAudio configurations, the best performance was achieved when the reference video was provided. However, despite achieving commendable temporal alignment, the audio generated by MMAudio displayed limited similarity to the reference audio, thereby validating our framework's design and training strategy. Finally, we tested audio dubbing on 6-second videos using 2 seconds of reference audio from the Greatest Hits dataset, with results from MMAudio and FoleyGenEx available on our website for material transfer comparison, and we also showcase FoleyGenEx's transfer results with semantically irrelevant audio events.

## 5.4 FE

For the FE task, we adhered to MultiFoley's protocol, randomly selecting 1,000 8-second clips from the VGGSound-Sync test set (Chen et al., 2021). Our aim was to generate audio for the 3–8 second segment (totaling 5 seconds) of each clip. We assessed performance across four conditions (Table 5), all derived from the same 8-second test clip: $V_t$ (Target video: 3–8 seconds), Text (Caption semantically matching the target video), $A_r$ (Reference audio: first 3 seconds), and $V_r$ (Reference video: first 3 seconds). To mitigate any bias from random sampling, we also present results using the complete test set. As demonstrated in Table 5, our method surpasses MultiFoley, particularly in temporal alignment. Although MMAudio incorporates a robust temporal alignment module, its lack of a reference audio conditional injection design hinders its ability to achieve style continuity. This deficiency creates a discrepancy between the training and inference phases, making it challenging to generate audio that aligns with the conditional style, ultimately resulting in low similarity to the ground truth.

## 5.5 ADVERB AUGMENTATION

To further assess the effectiveness of our adverb-based data augmentation algorithm in improving fine-grained textual control, we integrated the generated adverb-augmented data into MMAudio's training set and retrained the model. For evaluation, we selected 122 samples containing adverbs from the AudioCaps test set. Table 6 shows the performance of MMAudio and our models on this subset, both before and after retraining with the augmented data. The results indicate that incorporating the augmented data leads to enhanced performance. Additionally, we carefully curated 50 adverb-containing captions specifically for subjective evaluation (captions will be made public). We

---

[7]https://github.com/resemble-ai/Resemblyzer

performed a subjective Good, Same, Bad (GSB) evaluation, comparing samples generated by the original pre-trained MMAudio model with those generated by the retrained model using our augmented data. Details of the GSB evaluation are available in Appendix C. The enhanced MMAudio achieved a G:S:B ratio of 386:66:48, resulting in a score of 3.965. This suggests that in 386 evaluation instances, the audio generated by the enhanced MMAudio was considered more consistent with the caption compared to the audio generated by the original MMAudio.

## 6 CONCLUSIONS

We introduces FoleyGenEx, an MMDiT-based VTA framework that addresses key limitations of existing methods by separating semantic and synchronization information processing. Equipped with a conditional injection mechanism for audio-guided tasks, a multi-modal dynamic masking strategy for stable temporal alignment, and an adverb-based data augmentation algorithm for fine-grained control, FoleyGenEx unifies six core capabilities (TTA, VTA, TC-VTA, AC-VTA, FE, and audio editing). Experiments on multiple datasets confirm it retains MMAudio's synchronization strength, extends MultiFoley's control versatility, and adds unique adverb-level precision, outperforming baselines across critical metrics to advance controllable VTA generation.

## 7 ETHICS STATEMENT

FoleyGenEx is a research project dedicated to advancing multi-modal controlled VTA generation, addressing limitations like audio-visual synchronization, multi-modal control, and semantic precision found in current methods. While offering benefits such as restoring audio for silent films, automating Foley sound design, and democratizing audio creation for those with limited expertise, it also presents risks of misuse. The ability to generate realistic, synchronized audio could be exploited for deceptive media. The fine-grained editing function could also be misused to alter audio context where authenticity is crucial, such as legal or journalistic settings. To counter these risks, we stress that FoleyGenEx users must follow ethical and legal guidelines. The model should not be used to infringe on privacy, impersonate without consent, or spread misinformation.

## 8 REPRODUCIBILITY STATEMENT

To ensure the reproducibility of FoleyGenEx's results, we've thoroughly documented data processing, model configurations, and experimental details. First, Section 4.1 specifies training data composition, with Appendix B detailing data preprocessing steps for precise replication. Second, Section 3.1 details FoleyGenEx's core components, and Section 4.2 specifies key training parameters. Appendix A describes the masked mean squared error loss design, essential for model reconstruction. Third, Sections 5.1–5.5 outline test sets and evaluation metrics, with clear inference settings. Finally, the model implementation is based on the open-source MMAudio framework[8], using the MMAudio-L-44.1kHz configuration to facilitate codebase adaptation for experiment reproduction.

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

## A  Loss Function Design

We utilize a masked mean squared error (MSE) loss, denoted as $L_{\mathrm{MSE_{Masked}}}$, which focuses back-propagation solely on the masked portions of each sample. The original MSE loss in MMAudio is defined as:

$$L_{\mathrm{MSE_{MMAudio}}} = \frac{1}{\mathrm{B} \times \mathrm{T} \times \mathrm{F}} \sum_{\mathrm{b=1}}^{\mathrm{B}} \sum_{\mathrm{t=1}}^{\mathrm{T}} \sum_{\mathrm{f=1}}^{\mathrm{F}} \left( \hat{\mathrm{y}}_{\mathrm{b,t,f}} - \mathrm{y}_{\mathrm{b,t,f}} \right)^2,$$

where $\hat{\mathrm{y}}_{\mathrm{b,t,f}}$ represents the model's prediction, $\mathrm{y}_{\mathrm{b,t,f}}$ is the ground truth, and B, T, and F denote the batch size, time dimension, and feature dimension, respectively. To concentrate on the masked regions, we first compute a frame-level loss for each time step:

$$L_{\mathrm{MSE_{Frame}}(b,t)} = \frac{1}{\mathrm{F}} \sum_{\mathrm{f=1}}^{\mathrm{F}} \left( \hat{\mathrm{y}}_{\mathrm{b,t,f}} - \mathrm{y}_{\mathrm{b,t,f}} \right)^2.$$

Then, we extract the masked frame losses using a boolean matrix $\mathrm{rand\_span\_mask}(b, t)$. This matrix indicates whether the frame at the $t$-th time step of the $b$-th sample is part of a masked span. The final masked loss is calculated by averaging these selected frame losses:

$$L_{\mathrm{MSE_{Masked}}} = \frac{1}{N} \sum_{\mathrm{b=1}}^{\mathrm{B}} \sum_{\mathrm{t=1}}^{\mathrm{T}} \left( L_{\mathrm{MSE_{Frame}}(b,t)} \times \mathrm{rand\_span\_mask}(b, t) \right),$$

where $N$ is the total number of masked frames across the batch ($N = \sum_{\mathrm{b=1}}^{\mathrm{B}} \sum_{\mathrm{t=1}}^{\mathrm{T}} \mathrm{rand\_span\_mask}(b, t)$), ensuring proper normalization by the number of masked elements.

## B  Dataset Processing

For VGGSound, all videos were truncated to 8 seconds, using the final segment to remove irrelevant leading content. The dataset's built-in category labels were used as text input. For AudioCaps, entries without visual components were excluded, and audio was truncated to 8 seconds. WavCaps data

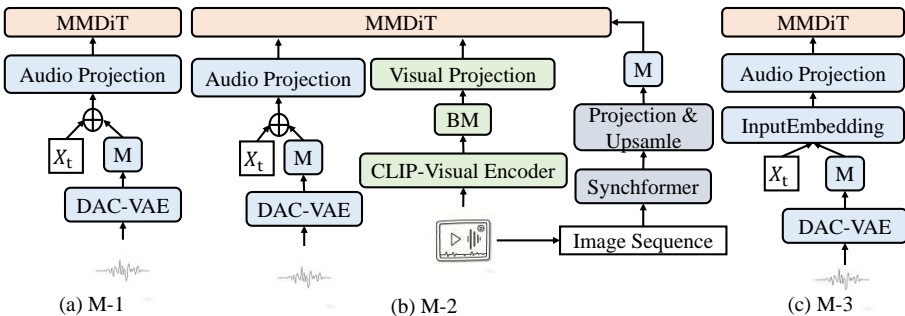

(a) M-1    (b) M-2    (c) M-3

Figure 5: Different mask training strategies: (a) Masking on the audio branch (b) Multimodal masking (c) Masking on the audio branch with InputEmbedding module.

Table 7: Comparison of audio-controlled video-to-audio performance across different mask training strategies.

| Method | OnsetSyncAP(%) ↑ | FD$_{VGG}$ ↓ | Resemblyzer ↑ | CLAP$_A$ ↑ |
|---|---|---|---|---|
| M-1 | 66.97 | 3.45 | 0.8430 | 0.5141 |
| M-2 | 68.90 | 2.91 | 0.8470 | 0.5224 |
| M-3 | 67.95 | 3.13 | 0.8677 | 0.7202 |
| FoleyGenEx (w/o summation) | 69.26 | 0.65 | 0.8562 | 0.7106 |
| FoleyGenEx | **69.38** | **0.54** | **0.9085** | **0.7216** |

was processed as follows: (1) audio clips shorter than 16 seconds were truncated to 8 seconds; (2) for longer clips (16 seconds or more), up to five non-overlapping 8-second segments were extracted to minimize redundancy. Samples with invalid text were also removed.

Our adverb-augmented data was generated in a multi-stage process. Initially, FFmpeg removed leading and trailing silence from all audio samples. Subsequently, specific audio augmentations were applied, guided by the adverbs within each sample's original caption: SoX handled speed augmentation, Pyroomacoustics simulated distance augmentation by adding room reverberation, and FFmpeg implemented gradual dynamic volume augmentation. Finally, captions were generated and refined using GPT-4.

To improve training efficiency, we precomputed and saved audio latent vectors, visual features, and text features as binary files, enabling direct loading during training and avoiding on-the-fly computation. We maintained a balanced learning process by sampling VTA and TTA data in a 1:1 ratio.

## C  SUBJECTIVE GSB EVALUATION DETAILS

To evaluate semantic consistency with text, particularly concerning adverbs, we conducted a Good, Same, Bad (GSB) evaluation, recruiting 10 evaluators. Each evaluator was presented with two audio clips generated from the same input text, with the models' identities (pre-trained MMAudio vs. model retrained with adverb-augmented data) concealed. Evaluators assessed the second audio clip relative to the first based on the following criteria: (1) Good (G): The second audio was more semantically consistent with the text, especially regarding the adverbs. (2) Same (S): The two audio clips were indistinguishable. (3) Bad (B): The second audio was inferior to the first. To mitigate potential order biases, we randomized the presentation order for half of the evaluators. In total, 50 test samples were assessed, resulting in 500 evaluations (10 evaluators × 50 samples). The GSB ratio was derived from these evaluations, and the final score was calculated as: Score = (G + S) / (S + B).

## D  ABLATION EXPERIMENT

We evaluated three distinct mask training strategies (Figure 5) to assess the contributions of our proposed conditional injection mechanism and multimodal dynamic mask training strategy. This evaluation covered the AC-VTA task (Table 7) and the FE task (Table 8), examining: (a) audio-only

Table 8: Comparison of Foley extension performance. $C_A$:$CLAP_A$.

| Conditions | | | | MMAudio | | FoleyGenEx | | M-1 | | M-2 | | M-3 | |
|---|---|---|---|---|---|---|---|---|---|---|---|---|---|
| $V_t$ | Text | $A_r$ | $V_r$ | $C_A \uparrow$ | DeSync $\downarrow$ | $C_A \uparrow$ | DeSync $\downarrow$ | $C_A \uparrow$ | DeSync $\downarrow$ | $C_A \uparrow$ | DeSync $\downarrow$ | $C_A \uparrow$ | DeSync $\downarrow$ |
| ✓ | ✓ | | | 56.0 | 0.40 | 59.7 | 0.39 | 57.5 | 0.40 | 57.5 | 0.40 | 58.9 | 0.39 |
| ✓ | | ✓ | | 57.0 | 0.46 | 61.8 | 0.40 | 58.1 | 0.46 | 58.2 | 0.44 | 60.9 | 0.42 |
| ✓ | | ✓ | ✓ | 59.0 | 0.39 | 69.3 | 0.37 | 65.9 | 0.39 | 66.2 | 0.38 | 68.7 | 0.38 |
| ✓ | ✓ | ✓ | ✓ | 60.7 | 0.38 | 71.2 | 0.36 | 70.0 | 0.37 | 70.2 | 0.37 | 70.9 | 0.36 |

Figure 6: The inference setup for the audio-controlled video-to-audio task.

branch masking (Figure 5a); (b) multimodal masking (Figure 5b); and (c) audio branch masking with an InputEmbedding module for reference audio condition injection (Figure 5c).

For AC-VTA (Table 7), multimodal masking consistently surpassed audio-only masking strategies—even with the InputEmbedding module present—in audio-video synchronization (OnsetSyncAP). This is attributed to our multimodal strategy's ability to train the model to disentangle the reference audio from the synthetic reference video. Conversely, audio-only masking during training, where conditional audio is aligned to corresponding video segments, can lead the model to incorrectly assume this alignment persists with randomly cropped target video during inference. This misalignment hinders the model's ability to maintain synchronization, performing even worse than MMAudio under identical test conditions (MMAudio-S2 in Table 4).

Regarding audio style transfer, all three strategies improved both timbre and semantics over the original MMAudio (MMAudio-S2). Crucially, the InputEmbedding module enabled performance exceeding MMAudio's ideal setup (MMAudio-S3, with reference video), significantly boosting semantic metrics ($CLAP_A$).

For the FE task (Table 8), alignment metrics remained unaffected across all strategies due to the inherent alignment of reference and target audio. The InputEmbedding module still offered a slight overall performance improvement. In terms of style continuation, strategies without the InputEmbedding module already outperformed MMAudio, and its incorporation—specifically designed for style transfer—further enhanced this performance.

# E  COMPARISON WITH EXISTING CONDITIONAL INJECTION METHODS

## E.1  CORE ARCHITECTURAL DESCRIPTION OF OUR CONDITIONAL INJECTION MECHANISM

Integrated into FoleyGenEx's audio branch for reference audio-guided tasks, its key workflow is as follows:

- Latent Processing: Audio latents are extracted using a DAC-VAE. During training, 70–100% of these latents are randomly masked to simulate the variable duration of reference audio encountered during inference.

- Input Embedding Fusion: These masked latents are then concatenated with the flow-matching intermediate state $X_t$ along the channel dimension. Following a linear projection and ConvPositionEmbedding, the resulting fused features are input into the MMDiT model.

- Inference Injection: As illustrated in Figure 6, reference audio latents are integrated with initial noise through both concatenation and summation. Additionally, a "fake reference video"—generated by cropping or duplicating the target video to match the reference audio duration—serves as a placeholder. This placeholder provides synthetic synchronization features, which are then prepended to the target video's synchronization features to prevent misalignment.

## E.2  KEY DIFFERENCES FROM MAINSTREAM CONDITIONING METHODS

Table 9 presents a comparison of mainstream conditioning methods, evaluating their fusion logic, train-inference consistency, and temporal alignment.

| Dimension | FilM Layers | Cross-Attention (**MultiFoley**) | Summation with Latent Variables (**M-1**) | Our Conditional Injection (**FoleyGenEx**) |
|---|---|---|---|---|
| Fusion Logic | Feature affine transform (scale/shift) | Bidirectional feature interaction | Summation of latent variables | Concatenation (along the channel dimension) + summation + InputEmbedding module |
| Train-Inference Consistency | Poor zero-shot performance | Relies on full-modal interaction; poor generalization | No multimodal alignment consideration | Masked multi-modal training + conditional injection |
| Temporal Alignment | Prone to misalignment | Low alignment precision | Low alignment precision | Concatenation + summation + Fake reference video |

Table 9: Comparison with Mainstream Conditioning Methods

## E.3  EXPERIMENTAL VALIDATION OF ADVANTAGES

- FE Task: Tables 5 and 8 demonstrate that our FoleyGenEx surpasses MultiFoley (Cross-Attention) and M-1 (Summation with Latent Variables) in both audio-visual synchronization and style continuity.

- Ablation Studies: Ablation studies were conducted on the components of our conditional injection mechanism, specifically investigating three masking training strategies (Figure 5) and comparing their performance on the AC-VTA and FE tasks. Results, detailed in Tables 7 and 8, indicate that the multimodal masking training strategy maintains train-inference consistency and improves audio-visual synchronization. Furthermore, the conditional injection design of the audio branch effectively enhances style transfer.

## F  QUALITATIVE EVALUATION AGAINST MULTIFOLEY

We evaluated FoleyGenEx on 13 samples sourced from MultiFoley's demo page; the generated results are accessible on our demo page. Table 3 presents an objective metric comparison, revealing that our model significantly outperforms MultiFoley in audio-visual synchronization, specifically by reducing latency by over 100 milliseconds.

To further illustrate this improvement, Figure 7 compares the spectrograms of audio generated by both models for the same video under varying text inputs. The left panel of Figure 7 shows a video frame at 1.804s, depicting a bird with an open beak, indicating a vocalization action. Critically, MultiFoley's generated audio produces this vocalization only when the input text is 'sleep bleating' at the corresponding timestamp. In contrast, our model accurately captures this vocalization, generating perfectly timed audio irrespective of the text input.

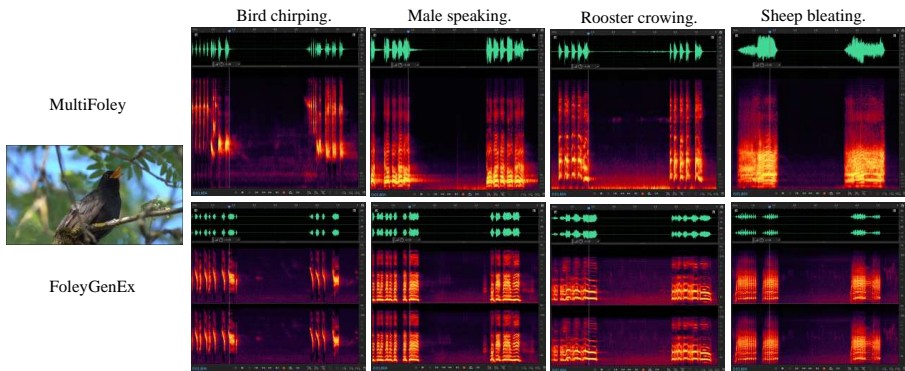

Figure 7: Spectrograms of audio generated by MultiFoley and our FoleyGenEx, using identical text and video inputs in the TC-VTA task.

# G   THE USE OF LARGE LANGUAGE MODELS (LLMS)

During the preparation of this paper, LLMs were used as a tool for writing assistance and polishing. Specifically, LLMs helped refine language, improve clarity and fluency. All content generated or refined by LLMs was carefully reviewed and verified by the authors to ensure accuracy, originality, and compliance with academic integrity. The authors take full responsibility for the paper's final content, including any portions that utilized LLMs.

