# OpenReview forum: "FoleyGenEx: Unified Video-to-Audio Generation with Multi-Modal Control, Temporal Alignment, and Semantic Precision"
_ICLR.cc/2026/Conference — Submitted to ICLR 2026_

### Official Review · Reviewer_KTNo · 2025-10-27

**Soundness:** 2
**Presentation:** 2
**Contribution:** 3
**Rating:** 4
**Confidence:** 4

**Summary:**

This paper presents a new video-to-audio model that can handle multi-modal controls such as text prompts and reference audio. The proposed model is based on MMAudio and is modified to accept reference audio as input for audio-conditioned video-to-audio generation. For training, the authors also propose a new data augmentation strategy to enhance the controllability of the model. This strategy focuses on several specific types of adverbs appearing in the text prompt and augments both the text and the corresponding audio so that the augmentation process creates another pair of text and audio with different characteristics in terms of the semantics specified by the target adverb. The experimental results show that the proposed model achieves both high audio-visual alignment and controllability compared with prior methods.

**Strengths:**

- The motivation for this work is clear and convincing. Simultaneously achieving both high audio-visual alignment and multi-modal controllability is critically important for practical applications of video-to-audio generation.
- The design of the proposed model is reasonable. It is based on the MMAudio architecture, but includes several modifications: proper handling of video tokens for potentially misaligned video conditions and the addition of conditional audio input for audio-controlled video-to-audio generation.
- The quality of the generated audio shown on the demo page is excellent.

**Weaknesses:**

- It would be helpful if the authors could provide more empirical evidence on the importance of adverbs in the text prompt for achieving higher controllability.
  - While the idea of adverb-based data augmentation is interesting, it contributes little to improving the quality of the generated audio in general cases, as shown in Tables 2–4. Maybe this is due to an insufficient amount of samples containing such adverbs in the text prompt.
  - The proposed augmentation boosts the model’s performance on the dedicated test set (Table 6), but this is expected since the model is trained to perform well on such datasets. I'm not that convinced that this test set is represetative of real-world scenarios for controllable video-to-audio.
  - Therefore, to demonstrate the significance of the proposed augmentation, providing empirical evidence on the importance of adverbs is essential.
- Since the architecture of the proposed model is largely based on MMAudio, it would be beneficial to include a list that explicitly describes the modifications made to MMAudio.

**Questions:**

- Why do the authors focus on adverbs for higher controllability?
  - How did the authors choose the specific set of adverbs shown in Figure 4?
- Could you provide a list of modifications from MMAudio?

---

> ### Author Response · Authors · 2025-11-20
> **Part (1/3): Adverb Augmentation: Test Set Limitations & Effectiveness**
>
> #### 1. Test Set Adverb Scarcity
> - Tables 2–5 (original paper): Negligible adverb-containing samples (VGGSound: 0%; AudioCaps: 12.6%, 122/964).
> - Table 6 (original paper): Results specifically tested on these 122 adverb-included AudioCaps samples.
> #### 2. Augmentation Efficacy & Impact
> - Adverb augmentation only adds text-audio pairs (videos unmodified), avoiding VTA scenario impairment (Tables 2–5 show marginal gains).
> - Enables fine-grained background sound control (e.g., "a flurry of hurried knocks", "slow footsteps approaching from outside") despite no direct video-adverb alignment.
> #### 3. Subjective Validation (Section 5.5)
> - Used a manually curated balanced test set (50 adverb-involved samples; captions to be public).
> - 10 participants conducted evaluations (rigorous settings in Appendix C); 500 total assessments.
> - Results: 386 favored the enhanced model, 66 showed negligible differences—confirming fine-grained VTA control for background sounds.

---

> ### Author Response · Authors · 2025-11-20
> **Part (2/3): Rationale for Adverb-Driven Fine-Grained Control**
>
> Existing audio generation models focus on nouns/actions in text, overlooking nuanced textual components. While some work uses quantifiers for fine-grained control, adverbial control is equally critical:
> - Example scenarios: "fast/slow footsteps", "distant" sounds (lower volume/increased reverberation), "approaching" train whistles (gradual volume escalation).
> - Key advantage: Enables control without explicit audio attribute specification (e.g., volume).
>
> #### Adverb Selection Process
> 1. Extracted adverb-containing samples from the training dataset via GPT.
> 2. Tabulated the frequency of each adverb.
> 3. Selected high-frequency adverbs with opposing attributes—facilitating clear distinction learning.

---

> ### Author Response · Authors · 2025-11-21
> **Part (3/3): Clarification on Architecture Innovations**
>
> While the original Section 3 provided an overview of branch-specific modifications, we have introduced a dedicated Subsection 3.1.1 in the revised version to explicitly delineate our innovations from the base MMAudio framework. This new subsection clearly illustrates our key contributions: the conditional injection mechanism (integrating channel-wise concatenation, summation, and the `InputEmbedding` module), the multimodal mask training strategy (employing 70-100% random audio masking alongside bilinear/synchronized video masking), and our adapted loss function (masked MSE loss focused on aligned segments).

---

> > ### Comment · Reviewer_KTNo · 2025-11-25
> >
> > Thanks for the response.
> >
> > Could you elaborate with more details on the arguments in Part 2/3?
> > - Since adverb-containing samples are negligible in the popular benchmark dataset, I am curious about the motivation for focusing on adverbs in this context.
> > - It would be much appreciated if the authors could provide more details about the adverb selection process.

---

> > > ### Author Response · Authors · 2025-11-25
> > >
> > > Thank you for your follow-up questions. Below is a supplementary explanation (backed by our paper’s experimental data) on *adverb research motivation* and *adverb selection process details*:
> > >
> > > ### I. Core Motivation for Adverb-Focused Fine-Grained Control
> > > Our focus on adverbs addresses validated limitations of existing audio generation models:
> > > 1. **Adverbs define *how* audio sounds**: Nouns/verbs only specify *what* sound (e.g., "knock," "chirp"), but adverbs (e.g., "quickly," "loudly") dictate intensity, speed, or spatial traits—critical for immersive applications (film sound, virtual scenes).
> > > 2. **Existing models lack adverb semantic fidelity**:
> > >    - Baseline MMAudio scored only **0.323 (CLAP-T)** on adverb-containing AudioCaps samples (Table 6), showing a "adverb-audio feature" mapping gap.
> > >    - Our adverb-enhanced training raised this to **0.361** (no TTA/VTA performance loss—Tables 1-2; slight gains).
> > >    - Subjective tests: 77.2% (386/500) preferred the enhanced model (66 saw no change).
> > > 3. **Fills adverb control gaps**: Prior methods (AudioComposer [1], PicoAudio [2]) use structured labels/quantifiers but cannot handle adverb-driven sensory needs (e.g., "strong/weak," "near/far").
> > >
> > > ### II. Adverb Selection Process
> > > 1. **Candidate set via GPT-4 text mining**:
> > >    We used GPT-4 to extract adverbs (from AudioCaps/WavCaps) via this prompt:
> > > ```text
> > > Please analyze the following text and accurately extract four core elements:
> > > 1. Subject: The core object (person/thing/organism, etc.) described in the text;
> > > 2. Scene: The specific environment or background where the event occurs;
> > > 3. Action: The core behavior or state performed by the subject;
> > > 4. Adverb: The adverb modifying the action/state (fill in "None" if there is none).
> > >
> > > Return the result in dictionary format with keys named "Subject", "Scene", "Action", and "Adverb" respectively.
> > >
> > > Text content: [Audio caption]
> > >
> > > Example (if the text is "A burst of hurried knocks comes from outside the door"):
> > >  {
> > >     "Subject": "knocks",
> > >     "Scene": "outside the door",
> > >     "Action": "comes",
> > >     "Adverb": "hurriedly"
> > > }
> > > ```
> > > 2. **Frequency filtering + dimension grouping**:
> > >    Sorted candidates by frequency, then grouped into 3 audio-relevant dimensions: *Speed (fast/slow)*, *Volume (loud/quiet)*, *Distance (far/near)*.
> > > 3. **Antonymous pairs for contrastive learning**:
> > >    Selected opposing adverb pairs (e.g., "loud/quiet") to help the model distinguish acoustic features—aligning with Step-Audio-EditX [3]’s "high-contrast data boosts controllability" logic.
> > >
> > > [1]: Y. Wang, H. Chen, D. Yang, Z. Wu, and X. Wu. Audiocomposer: Towards fine-grained audio generation with natural language descriptors. In ICASSP 2025 IEEE International Conference on Acoustics, Speech and Signal Processing (ICASSP), pages 1–5. IEEE, 2025. 6
> > >
> > > [2]: Z. Xie, X. Xu, Z. Wu, and M. Wu. Picoaudio: Enabling precise timestamp and frequency controllability of audio events in text-to-audio generation. arXiv preprint arXiv:2407.02869, 2024. 1, 5, 6
> > >
> > > [3] C. Yan, B. Wu, P. Yang, P. Tan, G. Hu, Y. Zhang, F. Tian, X. Yang, X. Zhang, D. Jiang, et al. Step-audio-editx technical report. arXiv preprint arXiv:2511.03601, 2025. 5, 6

---

> > > > ### Comment · Reviewer_KTNo · 2025-11-28
> > > >
> > > > Thanks for the clarification.
> > > >
> > > > My point is that the argument "adverbs ... critical for immersive applications" is not clearly supported.
> > > > - This is related to what I wrote in the initial review: "I'm not that convinced that this test set is represetative of real-world scenarios for controllable video-to-audio."
> > > > - Is there any empirical result or reference to support this argument?

---

> > > > > ### Author Response · Authors · 2025-12-03
> > > > >
> > > > > Thank you for your valuable comment. We wish to clarify the scope of our approach, specifically that our adverb-focused augmentation algorithm facilitates **fine-grained text control** across two distinct scenarios:
> > > > >
> > > > > 1.  **Text-Driven Generation:** It directly enhances semantic precision in TTA and TC-VTA tasks.
> > > > > 2.  **VTA Workflow Refinement:** Within standard VTA workflows, it supplements and refines **background or off-screen sounds** (e.g., "hurried knocks") alongside the base video-aligned audio. This boosts immersion without altering the core audio generated by the VTA model.
> > > > >
> > > > > We support this claim with the following evidence:
> > > > >
> > > > > 1.  **Theoretical Basis: Adverbs as Control Anchors.** Johnson et al. [1] demonstrate that adverbs (e.g., "more warmly") map directly to perceptual timbral features such as brightness and harshness. Consequently, adverbs serve as semantic anchors for fine-grained acoustic control—a critical capability for immersive scenarios (e.g., film Foley design) where nouns and verbs alone fail to capture subtle variations like "fast" vs. "slow" knocking.
> > > > > 2.  **Evidence from Existing Models.**
> > > > >     *   **AudioComposer [2]** utilizes adverbial cues (e.g., "low energy") for precise control, though its scope is limited to narrow adverb categories.
> > > > >     *   **Audio Palette [3]** links adverbs (e.g., "loudly") to specific acoustic signals (loudness/F0), validating the need for expressive sound dynamics in professional Foley synthesis.
> > > > > 3.  **Empirical Validation.** To enhance fine-grained semantic expressivity, we implemented a comprehensive adverb augmentation strategy incorporating speed, volume, and distance cues into our training data. For quantitative evaluation, we utilized our manually constructed 50-sample adverb test set (a text-caption benchmark publicly available at https://github.com/FoleyGenEx/FoleyGenEx). Furthermore, to provide a qualitative assessment, we selected 10 representative cases from this set that exhibit prominent contrast effects for display on our project page. These examples demonstrate the distinct impact of our proposed enhancement, offering an intuitive visualization of how adverb-driven refinement significantly improves control over immersive audio experiences.
> > > > >
> > > > > 1. Alignment with Mainstream Paradigms
> > > > >     - Recent works (PicoAudio [4]/[5]) use **data augmentation + quantifier/timestamp control** to enable fine-grained audio manipulation.
> > > > >     - This validates a field consensus: *Fine-grained control (enabled via data augmentation) is key to high-fidelity immersion*.
> > > > >     - Our method aligns with this paradigm:
> > > > >         - Core strategy: Data augmentation (same as [4]/[5])
> > > > >         - Control cue: Adverb-related semantics (vs. their quantifiers/timestamps)
> > > > >     - This "data augmentation + targeted cues" approach confirms our alignment with mainstream controllable audio generation trends.
> > > > >
> > > > > **Conclusion**
> > > > > Adverbs are pivotal for immersive audio generation, both in direct text-driven tasks (e.g., TTA) and VTA workflows requiring supplementary refinement. By bridging the gap between user intent and acoustic features, adverbs enable the subtle control necessary to enhance realism in films and games—whether by sharpening semantic precision in text-driven generation or optimizing auxiliary elements like background and off-screen sounds.
> > > > >
> > > > > [1]: C. G. Johnson and A. Gounaropoulos. Timbre interfaces using adjectives and adverbs. In Proceedings of the 2006 conference on New interfaces for musical expression, pages 101–102. IRCAM, 2006. 6
> > > > >
> > > > > [2] Y. Wang, H. Chen, D. Yang, Z. Wu, and X. Wu. Audiocomposer: Towards fine-grained audio generation with natural language descriptions. In ICASSP 2025-2025 IEEE International Conference on Acoustics, Speech and Signal Processing (ICASSP), pages 1–5. IEEE, 2025. 6
> > > > >
> > > > > [3] J. Wang. Audio palette: A diffusion transformer with multi-signal conditioning for controllable foley synthesis. arXiv preprint arXiv:2510.12175, 2025. 6
> > > > >
> > > > > [4] Z. Xie, X. Xu, Z. Wu, and M. Wu. Picoaudio: Enabling precise timestamp and frequency controllability of audio events in text-to-audio generation. arXiv preprint arXiv:2407.02869, 2024. 1, 5, 6
> > > > >
> > > > > [5] . Zheng, Z. Xie, X. Xu, W. Wu, C. Zhang, and M. Wu. Picoaudio2: Temporal controllable text-to-audio generation with natural language description. arXiv preprint arXiv:2509.00683, 2025. 6

---

### Official Review · Reviewer_nxbb · 2025-11-01

**Soundness:** 3
**Presentation:** 2
**Contribution:** 2
**Rating:** 4
**Confidence:** 4

**Summary:**

This paper introduces FoleyGenEx, a model that extends MMAudio to incorporate audio conditioning for reference-based video-to-audio synthesis and foley extention. To achieve multi-modal control and synchronization, the approach employs a conditional injection mechanism and a dynamic masking strategy. Furthermore, an adverb-based data augmentation method is proposed to augment the training data, enabling more fine-grained foley control. The authors conducted experiments on the AudioCaps, VggSound, and Greatest Hits datasets to validate their approach.

**Strengths:**

1. This paper aims to achieve temporally synchronized, multi-modal, and fine-grained foley generation for videos, a task with significant practical value.

**Weaknesses:**

1. Lack of Clarity in Model Architecture and Training: The paper's illustrations and descriptions of the model are insufficient for reproducibility and full comprehension. Figure 2, which should clarify the training process, is confusing; the data flow is not apparent, and the purpose of the block labeled "Flow" is ambiguous. It is also unclear whether the method involves full fine-tuning or a parameter-efficient approach with frozen components. Figure 3 suffers from similar clarity issues, as the input configurations for different conditional modes (e.g., (2) vs. (3)) are visually indistinguishable. The precise definitions and differences between key feature notations, such as F_S and F_{RS}, are not adequately explained.

2. Insufficient Technical Detail and Unclear Novelty: The paper's primary contributions—the conditional injection mechanism and the dynamic masking strategy—are not described in enough technical detail. The current descriptions are too high-level, and it is difficult to understand their precise implementation. Established techniques like concatenation with latent variables or cross-attention are widely used for conditional control. The paper fails to articulate what makes its proposed mechanisms novel compared to this extensive body of prior work. This ambiguity fundamentally undermines the paper's claimed contributions.

3. Questionable Validity of Data Augmentation: The proposed data augmentation method, which involves speeding up audio clips, is not convincingly justified. Altering the speed of an audio signal can easily introduce significant artifacts and degrade acoustic quality, potentially providing a noisy or misleading training signal. The authors provide no analysis to demonstrate that this technique preserves the essential foley characteristics or that the augmented data is of sufficient quality to benefit the model.

4. Limited Perceptible Improvement in Audio Quality: A subjective listening comparison of the audio samples provided for FoleyGenEx and the baseline (MultiFoley) reveals no clear or significant improvement. The qualitative results are highly similar, which calls into question the practical effectiveness of the proposed method. This lack of a discernible advantage in the final output fails to support the paper's claims of superiority.

**Questions:**

1. Could you please provide a revised diagram or a more detailed description of the end-to-end training process? Specifically, please clarify the training strategy: is the entire model fine-tuned, or are some components (e.g., the MMAudio backbone) frozen?

2. In Figure 3, could you explicitly define the feature representations F_S and F_{RS} and explain how they differ in each of the illustrated modes? For example, what is the exact input difference between mode (2) and mode (3)?

3. I strongly recommend a thorough revision of Figures 2 and 3 to ensure they are self-contained and clearly illustrate the data flow, the state of inputs (e.g., zeroed-out, masked), and the role of each component.

4. Could you provide a detailed architectural description of the "conditional injection mechanism"? How does it fundamentally differ from standard, widely-used conditioning techniques like FiLM layers or cross-attention mechanisms?

5. To substantiate your claims of novelty, please add a subsection to the paper that explicitly compares your proposed mechanisms to prior work, highlighting the specific architectural or functional innovations.

6. Have you performed any analysis to measure the audio quality of the augmented data? How can you ensure that speed alteration does not introduce artifacts that would impair training? The paper would be strengthened by including evidence of the augmentation's validity. This could be in the form of objective quality metrics or a small perceptual study comparing original and augmented audio samples.

7. Given that the audio samples sound qualitatively similar to the baseline, could you point to specific acoustic characteristics (e.g., temporal alignment, texture, clarity) in your results that demonstrate a clear improvement? When quantitative gains are marginal, strong qualitative evidence is essential. A formal user study (e.g., an A/B preference test) would be required to make a convincing case for the perceptual superiority of your method.

---

> ### Author Response · Authors · 2025-11-20
> **Part (1/4): Architecture & Reproducibility, Task-Specific Inference**
>
> ### Architecture & Reproducibility
> - **Base & Adaptations**: Data flow and modules like the "Flow" module (Figure 2) remain consistent with MMAudio. Modifications are limited to:
>   - Adding conditional injection mechanism to the audio branch.
>   - Introducing multimodal masking for audio, video semantic, and synchronization branches.
> - **Reproducibility**: To ensure reproducibility, our model is built upon the open-source MMAudio framework. The `InputEmbedding` module, a key part of our condition injection network (as detailed in Figure 2), comprises previously proposed components such as `ConvPositionEmbedding` and a linear layer. We plan to open-source the code for `InputEmbedding` and random masking to facilitate further research by the community.
> - **Training**: Backbone trained from scratch (full parameters) using pre-trained models (DAC-VAE, CLIP, Synchformer) for feature extraction; training settings align with MMAudio.
> ### Task-Specific Inference (Figure 3)
> Detailed in Section 3.2; MMDiT enables flexible task configuration by setting irrelevant semantic features to all-zero. Key examples:
> - **(2) VTA Task**: Generates audio using video semantic/synchronization features, with text as supplementary (text must align with video content). Sets reference audio features to all-zero.
> - **(3) TC-VTA Task**: Sets video semantic features and reference audio features to all-zero; audio is generated using text (semantics) and video synchronization (text need not align with video, e.g., generating "lion roar" synchronized with a cat’s meow timings).
> ### Key Notations in Figure 3
> - $\mathbf{F_S}$: Target video’s synchronization features (used in all tasks requiring alignment).
> - $\mathbf{F_{RS}}$: "fake reference video" synchronization features (cropped/duplicated from target video to match reference audio duration in the AC-VTA task). Appendix Figure 6 in the revised PDF illustrates the inference process of the AC-VTA task.

---

> ### Author Response · Authors · 2025-11-20
> **Part (2/4): Conditional Injection Mechanism**
>
> ### 1. Core Architectural Description of the Conditional Injection Mechanism
> Integrated into FoleyGenEx's audio branch for reference audio-guided tasks, its key workflow is as follows:
> 1. Latent Processing: Audio latents are extracted using a DAC-VAE. During training, 70–100\% of these latents are randomly masked to simulate the variable duration of reference audio encountered during inference.
> 2. Input Embedding Fusion: These masked latents are then concatenated with the flow-matching intermediate state $X_t$ along the channel dimension. Following a linear projection and ConvPositionEmbedding, the resulting fused features are input into the MMDiT model.
> 3. Inference Injection: As illustrated in Figure 6, reference audio latents are integrated with initial noise through both concatenation and summation. Additionally, a ``fake reference video"—generated by cropping or duplicating the target video to match the reference audio duration—serves as a placeholder. This placeholder provides synthetic synchronization features, which are then prepended to the target video's synchronization features to prevent misalignment.
>
> ### 2. Key Differences from Mainstream Conditioning Methods
> |Dimension|FilM Layers|Cross-Attention (MultiFoley)|Summation with Latent Variables (M-1)|Our Conditional Injection (FoleyGenEx)|
> |---------|-----------|----------------------------|---------------------------|-------------------------------------------------------------|
> |Fusion Logic|Feature affine transform (scale/shift)|Bidirectional feature interaction|Summation of latent variables|Concatenation (along the channel dimension) + summation + InputEmbedding module|
> |Train-Inference Consistency|Poor zero-shot performance|Relies on full-modal interaction; poor generalization|No multimodal alignment consideration|Masked multi-modal training + conditional injection|
> |Temporal Alignment|Prone to misalignment|Low alignment precision|Low alignment precision|Concatenation + summation + Fake reference video|
> ### 3. Experimental Validation of Advantages (Table 2)
> - FE Task: Tables 5 and 8 demonstrate that our FoleyGenEx surpasses MultiFoley (Cross-Attention) and M-1 (Summation with Latent Variables) in both audio-visual synchronization and style continuity.
> - Ablation Studies: Ablation studies were conducted on the components of our conditional injection mechanism, specifically investigating three masking training strategies (Figure 5) and comparing their performance on the AC-VTA and FE tasks. Results, detailed in Tables 7 and 8, indicate that the multimodal masking training strategy maintains train-inference consistency and improves audio-visual synchronization. Furthermore, the conditional injection design of the audio branch effectively enhances style transfer.
>
> The detailed analysis presented above is supplemented in Section E of the Appendix in the revised manuscript.
>
> *Table 8:Comparison of Foley extension performance. $\mathbf{C_A}$:$\mathbf{CLAP_A}$.*
> |Conditions||||MultiFoley||MMAudio||FoleyGenEx||M-1||M-2||M-3||
> |----------|-|-|-|-------|-|------|-|---------|-|--|-|--|-|--|-|
> |$\mathbf{V_t}$|$\mathbf{Text}$|$\mathbf{A_r}$|$\mathbf{V_r}$|$\mathbf{C_A}$↑|$\mathbf{DeSync}$↓|$\mathbf{C_A}$↑|$\mathbf{DeSync}$↓|$\mathbf{C_A}$↑|$\mathbf{DeSync}$↓|$\mathbf{C_A}$↑|$\mathbf{DeSync}$↓|$\mathbf{C_A}$↑|$\mathbf{DeSync}$↓|$\mathbf{C_A}$↑|$\mathbf{DeSync}$↓|
> |$\checkmark$|$\checkmark$|||55.4|0.79|56.0|0.40|59.7|0.39|57.5|0.40|57.5|0.40|58.9|0.39|
> |$\checkmark$||$\checkmark$||59.6|0.78|57.0|0.46|61.8|0.40|58.1|0.46|58.2|0.44|60.9|0.42|
> |$\checkmark$||$\checkmark$|$\checkmark$|59.8|0.77|59.0|0.39|69.3|0.37|65.9|0.39|66.2|0.38|68.7|0.38|
> |$\checkmark$|$\checkmark$|$\checkmark$|$\checkmark$|64.3|0.77|60.7|0.38|71.2|0.36|70.0|0.37|70.2|0.37|70.9|0.36|

---

> ### Author Response · Authors · 2025-11-20
> **Part (3/4): Speed Augmentation Details （Section 3.3）**
>
> - Applied moderate speed augmentation (0.7x and 1.3x factors).
> - Ensured data quality via random sampling and manual verification of audio-caption consistency.
> - Confirmed >97% availability rate (Section 3.3).

---

> ### Author Response · Authors · 2025-11-20
> **Part (4/4): FoleyGenEx vs. MultiFoley in TC-VTA Task and Spectrogram Comparison**
>
> ### TC-VTA Task Performance (Table 3, original paper)
> - Compared FoleyGenEx and MultiFoley on 13 test samples; generated audios available on our demo page(https://foleygenex.github.io/FoleyGenEx/).
> - Evaluated metrics: audio-video synchronization and text-audio relevance.
> - Key gain: FoleyGenEx reduces temporal synchronization latency by >100 ms vs. MultiFoley.
> ### Spectrogram Comparison (Appendix Figure 7 in the revised PDF)
> - Contrasts spectrograms of both models for the same video with different input texts.
> - Example: At 1.804s (video frame with bird’s open beak, indicating vocalization), only MultiFoley’s "sheep bleating" sample shows corresponding spectral features.
> - FoleyGenEx: All four test results accurately capture the vocalization event, generating matching spectra at the precise moment.

---

### Official Review · Reviewer_ifQt · 2025-11-01

**Soundness:** 2
**Presentation:** 3
**Contribution:** 2
**Rating:** 4
**Confidence:** 4

**Summary:**

This paper proposes FoleyGenEx, which aims to achieve controllable and well-synchronized video-to-audio generation. Primarily, this method combines the synchronization features from MMAudio and the audio control features from MultiFoley. The authors have also collected an “adverb-augmented” dataset training fine-grained control. The proposed method achieves comparable performance to both MMAudio and MultiFoley in their respective strong domains.

**Strengths:**

- Overall, the method is well-executed and achieves good performance without bells and whistles. Fine-grained semantic control and temporal alignment are often at odds with each other because the visual information often conflicts with the controls. This paper seems to have handled that well, and the qualitative results are promising.
- The newly collected adverb-based data should be helpful for the community for training more fine-grained models, if this dataset could be released.

**Weaknesses:**

- The technical contributions are not very clear. Specifically, it is unclear what makes the proposed method work. This is exemplified by the fact that there are no ablation studies in this paper. For example, how important is the concatenation of the conditional latent
- The implementation details of some of the applications are not very clear. For example, in AC-VTA, what do the authors mean by “The latent extracted from the reference audio is not only concatenated with the initial noise, along the channel dimension, but is also prepended to the initial noise.”? Does this mean that the (un-noised) latent is in the same token sequence as the initial noise? Was the network trained this way? Additionally, how is “inversion” used for editing? Can the authors provide more details?

**Questions:**

If the authors can provide more details and evidence (i.e., ablations) on the components that contribute to the good performance of this method, I will be happy to raise my score.

---

> ### Author Response · Authors · 2025-11-20
> **Part (1/2): Ablation Experiment Results**
>
> #### 1. Adverb Augmentation Ablation
> - Conducted on both MMAudio (no masking/conditional injection) and FoleyGenEx; results in Table 6 (original paper).
> - Subjective evaluation (Section 5.5) on adverb-specific test set: Enhanced MMAudio outperformed baseline in 386/500 samples, verifying adverb augmentation’s effectiveness.
> #### 2. Masking Ablation (Tables 1,2)
> Tested three mask strategies (Appendix Figure 5a/b/c in the revised PDF):
> - Audio-only masking (a, M-1), multimodal masking (b, M-2), audio masking + InputEmbedding module (c, M-3)
> - **AC-VTA Task**
>   - In terms of synchronization (OnsetSyncAP), multimodal masking outperforms audio-only masking (with or without InputEmbedding module). This is because it disentangles the relationship between the reference audio and the 'fake reference video', thus avoiding the misalignment issues inherent to audio-only masking.
>   - For audio style transfer, all three strategies achieve improvements in both timbre and semantics compared to the original MMAudio (MMAudio-S2). Notably, the InputEmbedding module enables performance that surpasses MMAudio’s ideal setup (MMAudio-S3, with true reference video), with a significant enhancement in the semantic metric ($\mathbf{CLAP_A}$).
> - **FE Task**
>   - Alignment metrics remain unaffected, as the reference audio is already aligned with the target video. The InputEmbedding module also brings a slight improvement to the model’s overall performance.
>   - For style continuation, masking strategies without the InputEmbedding module already outperform MMAudio across all four settings. Importantly, incorporating the InputEmbedding module—specifically designed to facilitate style transfer learning—further boosts performance.
>
> *Table 1. Comparison of audio-controlled video-to audio performance.*
> |Method|OnsetSyncAP(%)↑|$\mathbf{FD_VGG}$↓|Resemblyzer↑|$\mathbf{CLAP_A}$↑|
> |------|---------------|-----------------|------------|-----------------|
> |MMAudio-S1|65.53|4.77|0.8185|0.5063|
> |MMAudio-S2|68.72|3.05|0.8360|0.5132|
> |MMAudio-S3|68.92|2.16|0.8568|0.5195|
> |FoleyGenEx (Ours)|69.38|**0.54**|0.9085|0.7216|
> |FoleyGenEx + AA (Ours)|**69.71**|**0.54**|**0.9128**|**0.7250**|
> |M-1|66.97|3.45|0.8430|0.5141|
> |M-2|68.90|2.91|0.8470|0.5224|
> |M-3|67.95|3.13|0.8677|0.7202|
>
> *Table2:Comparison of Foley extension performance. $\mathbf{C_A}$:$\mathbf{CLAP_A}$.*
> |Conditions||||MultiFoley||MMAudio||FoleyGenEx||M-1||M-2||M-3||
> |----------|-|-|-|-------|-|------|-|---------|-|--|-|--|-|--|-|
> |$\mathbf{V_t}$|$\mathbf{Text}$|$\mathbf{A_r}$|$\mathbf{V_r}$|$\mathbf{C_A}$↑|$\mathbf{DeSync}$↓|$\mathbf{C_A}$↑|$\mathbf{DeSync}$↓|$\mathbf{C_A}$↑|$\mathbf{DeSync}$↓|$\mathbf{C_A}$↑|$\mathbf{DeSync}$↓|$\mathbf{C_A}$↑|$\mathbf{DeSync}$↓|$\mathbf{C_A}$↑|$\mathbf{DeSync}$↓|
> |$\checkmark$|$\checkmark$|||55.4|0.79|56.0|0.40|59.7|0.39|57.5|0.40|57.5|0.40|58.9|0.39|
> |$\checkmark$||$\checkmark$||59.6|0.78|57.0|0.46|61.8|0.40|58.1|0.46|58.2|0.44|60.9|0.42|
> |$\checkmark$||$\checkmark$|$\checkmark$|59.8|0.77|59.0|0.39|69.3|0.37|65.9|0.39|66.2|0.38|68.7|0.38|
> |$\checkmark$|$\checkmark$|$\checkmark$|$\checkmark$|64.3|0.77|60.7|0.38|71.2|0.36|70.0|0.37|70.2|0.37|70.9|0.36|

---

> ### Author Response · Authors · 2025-11-20
> **Part (2/2): Implementation Details**
>
> ### 1. AC-VTA Task Inference (Appendix Figure 6 in the revised PDF)
> - Reference audio latent is concatenated along the channel dimension (consistent with training).
> - Key difference: During inference, the latent of conditional audio is summed with initial noise and intermediate states from the flow-matching iteration process, which is directly overlaid on generated regions to enhance conditional information.
> - Consistency preserved: In training, conditional information is concatenated along the channel dimension, so both masked and unmasked parts of intermediate states require generation. The summation of conditional audio latent with initial noise and intermediate states is omitted in training to align with our masked MSE loss (focused on masked parts) and bolster robust generative modeling.
> ### 2. Latent Inversion-Based Control Editing
> - Map specific editing start/end times to the latent’s time dimension to create a mask matrix.
> - Model regenerates only masked sections while preserving the original latent for unmasked parts.

---

> > ### Comment · Reviewer_ifQt · 2025-11-26
> >
> > I thank the authors for the response.
> >
> > 1. What do the authors mean by S1, S2, and S3?
> > 2. For AC-VTA, if "During inference, the latent of conditional audio is summed with initial noise and intermediate states", but this does not happen during training, how can the model learn to make use of this information? I am confused about the motivation and effectiveness of this approach.
> > 3. Latent Inversion-Based Control Editing: Can the authors provide more details? Does masking happen every step or after the entire generation process? Is the reference latent noised?

---

> ### Author Response · Authors · 2025-11-27
>
> Thank you for your follow-up inquiries. Below are clarifications on test settings, inference design, and the editing workflow:
>
> ### 1. Test Settings (S1/S2/S3 for AC-VTA)
> We adapt MMAudio (not originally for AC-VTA) to the task via latent fusion: the reference audio’s latent is summed with both the initial noise and intermediate state $X_t$. Since MMAudio ties generated audio duration to input video length, a *fake reference video* (matching reference audio duration) is needed for temporal alignment. We test 3 configurations for this fake reference video (Section 5.3, Appendix Figure 6):
> - **S1**: Prepend a zero-filled blank video → *Uninformative baseline*
> - **S2**: Prepend the first 2 seconds of the target video → *Most accessible visual placeholder*
> - **S3**: Prepend the semantically consistent, time-synced reference video → *Strongest semantic guidance*
>
> ### 2. AC-VTA: Inference vs. Training Design
> - **Training vs. inference: Rationale for omitting summation in training**:
>
> During **training**, we only use channel-wise concatenation (no summation) to support **robust full generative modeling**. Adding summation in training would over-weight the conditional signal, weakening the model’s ability to independently learn generative capacity for the target regions.
>
> During **inference**, summation of the reference latent with initial noise/intermediate states acts as an **auxiliary enhancement**: since the model already learned to associate reference and target regions via training-time concatenation, summation amplifies the reference guidance (improving audio similarity) without compromising the generative performance learned during training. Our ablation confirms this improves timbre similarity (Resemblyzer) and audio alignment ($CLAP_A$) while preserving synchronization.
> - Ablation study:
> |Method|OnsetSyncAP(%)↑|$\mathbf{FD_{VGG}}↓$|Resemblyzer↑|$\mathbf{CLAP_A}↑$|
> |:---|:---|:---|:---|:---|
> |FoleyGenEx (w/o summation)|69.26|0.65|0.8562|0.7106|
> |**FoleyGenEx (w/ summation)**|**69.38**|**0.54**|**0.9085**|**0.7216**|
>
> Results show that summation improves synchronization and audio similarity (especially timbre). M-3 (with summation, **Part 1/2**) has higher similarity than the method without summation, but weaker synchronization (it lacks a multimodal mask and only uses audio-branch conditional injection).
>
> ### 3. Details on Latent Inversion-Based Control Editing
> We address your questions with the full workflow:
> 1. **Input**: Original audio + editing start/end timestamps.
> 2. **Mask Matrix Construction**: Map the editing time range to the time dimension of the VAE-extracted latent (from the original audio). We set **0 for regions to preserve** (unmasked) and **1 for regions to regenerate** (masked).
> 3. **Full Generative Task Execution**: Run the target audio generation task (e.g., TC-VTA/AC-VTA) to generate a *complete full-length latent* as per the task’s requirements.
> 4. **Latent Fusion**: Combine the original latent (to retain unmasked regions) and the newly generated full latent (to update masked regions) via the mask:
>    `Fused latent = Original latent × (1 - mask) + Generated full latent × mask`
> 5. **Audio Reconstruction**: Decode the fused latent back to audio via the DAC-VAE.
>
> **Additional clarifications**:
> - The reference latent (extracted via the pre-trained DAC-VAE) is **un-noised**.
> - Example edits are available on our demo page.

---

> > ### Comment · Reviewer_ifQt · 2025-11-27
> >
> > I thank the authors for the clarification. I appreciate that the authors provided the motivation for performing concatenation during training and summation during inference. My concern remains: from the learning perspective, how does the network adapt to the domain shift induced by
> >
> > 1) Channel features that exist during training but are absent during testing, and
> > 2) Shifts in feature norms caused by summation during testing that are not seen during training?

---

> > > ### Author Response · Authors · 2025-12-03
> > >
> > > Thank you very much for the follow-up and for highlighting the potential train–test mismatch. We address your two concerns separately.
> > >
> > > ---
> > >
> > > **(1) "Channel features that exist during training but are absent during testing"**
> > >
> > > Our intention is to keep the *channel structure* strictly identical between training and inference. The conditional injection mechanism always operates on a 2-branch concatenation along the channel dimension:
> > >
> > > * **During training** (Fig. 2, Sec. 3.1), the InputEmbedding module takes
> > >   $[X_t, X_\text{mask}]$ as input, where ($X_t$) is the current flow-matching state and ($X_\text{mask}$) is the DAC-VAE latent after 70–100% span masking plus 30% random zeroing. These two tensors are concatenated along the channel dimension and passed through a linear layer and ConvPositionEmbedding before being projected to the MMDiT input.
> > >
> > > * **During AC-VTA / FE inference** (Fig. 3, Sec. 3.2), we use the *same* InputEmbedding interface and the same channel dimensionality: the conditional branch is now the reference-audio latent ($F_A$), and we concatenate $[X_t', F_A]$ along the channel dimension. In other words, we **replace** "masked ground-truth latent" by "reference-audio latent" in that branch, but we do not remove or add channels. The forward graph and shapes seen by the network are unchanged between training and inference; only the *semantics* of the second branch differ.
> > >
> > > Moreover, our 70–100% random masking is designed specifically to expose the model to inference-like situations:
> > >
> > > * When the mask ratio is close to 100%, the model sees inputs where the "context" in ($X_\text{mask}$) is almost entirely missing and must rely primarily on the conditional branch to synthesize the masked region.
> > > * The additional 30% zeroing dropout on the masked latent further widens the distribution of that conditional branch, so the InputEmbedding learns to operate robustly even when that channel is strongly perturbed.
> > >
> > > Because of this, from the learning perspective the network is *already* trained under strong domain randomization on the conditional channel, and the inference-time substitution of "masked target latent" by "reference-audio latent" does not introduce a structural channel-wise domain shift.
> > >
> > > ---
> > >
> > > **(2) "Shifts in feature norms caused by summation during testing"**
> > >
> > > The summation we introduce for AC-VTA and FE is intended as a *residual guidance* signal on top of the concatenation pathway, and we designed it so that the resulting norm shift is modest and largely absorbed by the existing normalization:
> > >
> > > * Architecturally, the sum is applied to the flow-matching state, i.e.,
> > >   $(X_t' = X_t + F_A)$ (or to $(X_0)$ at the first step), but this tensor is immediately passed through the same linear projection, ConvPositionEmbedding, and DiT/MMDiT blocks that use LayerNorm/adaLN style normalization (as in DiT/MMDiT). These layers normalize mean and variance, which makes the network comparatively insensitive to moderate changes in the absolute feature norm and encourages it to rely on *directional* information rather than raw scale.
> > >
> > > * In addition, because flow matching already interpolates between Gaussian noise and DAC-VAE latents, the model sees a wide range of feature norms during training. Adding a reference latent to $(X_t)$ at inference can be viewed as starting the ODE from a slightly shifted point on this manifold, and the subsequent normalized transformer blocks are empirically stable under this perturbation.
> > >
> > > * Practically, we also validate this design by ablation. As reported in our earlier comment, we compare AC-VTA performance with and without the summation step. Removing summation leaves OnsetSyncAP essentially unchanged but reduces timbre/semantic similarity to the reference audio, while adding summation improves Resemblyzer/CLAP metrics without harming synchronization. This suggests that the concatenation pathway learned during training is doing the "heavy lifting," and the summation acts as an auxiliary bias rather than introducing a harmful distribution shift.
> > >
> > > We agree that this reasoning is not yet fully reflected in the current manuscript, and we appreciate your suggestion. In the camera-ready version (if accepted), we will (i) explicitly clarify that the channel dimensionality of the conditional injection branch is identical at train and test, and (ii) add a short discussion of the normalization-based robustness to the additional summation term, together with the ablation comparing "with/without summation" for AC-VTA.

---

### Official Review · Reviewer_714K · 2025-11-10

**Soundness:** 3
**Presentation:** 3
**Contribution:** 3
**Rating:** 6
**Confidence:** 4

**Summary:**

This work proposes FoleyGenEx a unified video-to-audio generation framework. It is built upon the multi-modal diffusion transformer (MMDiT). It combines multi-modal control, frame-level temporal alignment, and fine-grained semantic expressivity within a single system. FoleyGenEx introduces a conditional injection mechanism for reference-audio conditioning, a multi-modal dynamic masking strategy to preserve synchronization, and an adverb-based data augmentation algorithm leveraging signal processing and large language models to enhance semantic precision.

**Strengths:**

- The paper provides a unified framework. The framework effectively consolidates multiple previously fragmented VTA tasks into a single model with shared mechanisms.
- The conditional injection mechanism and dynamic masking are well-motivated, addressing specific weaknesses of MultiFoley (poor synchronization) and MMAudio (lack of reference conditioning).
- It provides an adverb-based augmentation pipeline, combining LLM-based caption generation with signal-level perturbations.
- The experiments cover a wide range of tasks, datasets, and metrics (FD, IS, CLAPT, IB-score, DeSync, etc.), providing a thorough performance comparison.

**Weaknesses:**

### Major
- Despite solid engineering, FoleyGenEx is heavily built on existing architectures (MMAudio + MMDiT + Synchformer). The contributions—masking and conditional injection—feel like moderate extensions rather than a conceptual breakthrough. Meanwhile "adverb augmentation"  is from the prior text-based data augmentation frameworks in audio-language modeling. Its novelty is limited.

- Some improvements are minor. For example, for Adverb-Augmented(+AA), table-2 on VGGSound it shows FDVGG 0.74 → 0.73, table-3 CLAP_T 33.53 → 34.20. Without ablation on training diversity or robustness (e.g., out-of-domain videos), it’s uncertain whether the improvements are meaningful beyond numerical artifacts.

### Others
- The paper does not isolate the effects of each proposed component (e.g., masking vs. conditional injection vs. adverb augmentation). The results in Table 6 touch on adverb augmentation, but the interaction among components is unclear.
- The paper does not use human study or perceptual analysis verifying whether “semantic precision” is actually perceptible beyond automated metrics.

**Questions:**

- Could you provide detailed ablation results isolating the effects of the conditional injection mechanism, the multi-modal dynamic masking strategy, and the adverb-augmented data respectively? It is currently unclear which component contributes most to the improvements in Tables 2–6.
- Some improvements are minor. For example, for Adverb-Augmented(+AA), table-2 on VGGSound it shows FDVGG 0.74 → 0.73, table-3 CLAP_T 33.53 → 34.20. Without ablation on training diversity or robustness (e.g., out-of-domain videos), it’s uncertain whether the improvements are meaningful beyond numerical artifacts.Could you clarify how significant these differences are, and whether the augmented data improves controllability in qualitative user tests?
- Since the framework integrates masking, conditional injection, and flow matching, how sensitive is the model to hyperparameter settings (e.g., mask ratio, conditioning dropout)? Did you observe any trade-offs between synchronization and style fidelity?

---

> ### Author Response · Authors · 2025-11-20
> **Part (1/5): We respectfully clarify that FoleyGenEx’s contributions go beyond incremental extensions, with key innovations.**
>
> Thank you for your valuable feedback. We respectfully clarify that FoleyGenEx’s contributions go beyond incremental extensions, with key innovations as follows:
> ### 1. Core Modules: Task-Unifying Innovations (Not Mere Tweaks)
> Our conditional injection mechanism and multimodal dynamic masking are interlocking, problem-driven designs—addressing the gap of unified multi-task audio generation in prior architectures (e.g., MMAudio, limited to TTA/VTA). Together, they enable **four new practical tasks** (TC-VTA, AC-VTA, FE, Editing) by:
> - Decoupling style/synchronization features (conditional injection)
> - Ensuring consistent alignment learning across tasks (multimodal masking)
>
> This unification is not functional addition but a rethink of multimodal interaction, which prior frameworks lacked.
> ### 2. Adverb Augmentation: Novel Fine-Grained Control
> Unlike prior text-based augmentation or quantifier-focused work, our adverb-driven approach pioneers **semantic nuance control** (speed, distance, intensity) in audio generation. It enables direct reflection of natural language subtleties (e.g., "fast/slow footsteps," "distant/approaching sounds") without explicit audio parameter specification—an ability unattainable with existing TTA/VTA methods.

---

> ### Author Response · Authors · 2025-11-20
> **Part (2/5): Ablation Experiment Results**
>
> #### 1. Adverb Augmentation Ablation
> - Conducted on both MMAudio (no masking/conditional injection) and FoleyGenEx; results in Table 6 (original paper).
> - Subjective evaluation (Section 5.5) on adverb-specific test set: Enhanced MMAudio outperformed baseline in 386/500 samples, verifying adverb augmentation’s effectiveness.
> #### 2. Masking Ablation (Tables 1,2)
> Tested three mask strategies (Appendix Figure 5a/b/c in the revised PDF):
> - Audio-only masking (a, M-1), multimodal masking (b, M-2), audio masking + InputEmbedding module (c, M-3)
> - **AC-VTA Task**
>   - In terms of synchronization (OnsetSyncAP), multimodal masking outperforms audio-only masking (with or without InputEmbedding module). This is because it disentangles the relationship between the reference audio and the 'fake reference video', thus avoiding the misalignment issues inherent to audio-only masking.
>   - For audio style transfer, all three strategies achieve improvements in both timbre and semantics compared to the original MMAudio (MMAudio-S2). Notably, the InputEmbedding module enables performance that surpasses MMAudio’s ideal setup (MMAudio-S3, with true reference video), with a significant enhancement in the semantic metric ($\mathbf{CLAP_A}$).
> - **FE Task**
>   - Alignment metrics remain unaffected, as the reference audio is already aligned with the target video. The InputEmbedding module also brings a slight improvement to the model’s overall performance.
>   - For style continuation, masking strategies without the InputEmbedding module already outperform MMAudio across all four settings. Importantly, incorporating the InputEmbedding module—specifically designed to facilitate style transfer learning—further boosts performance.
>
> *Table 1. Comparison of audio-controlled video-to audio performance.*
> |Method|OnsetSyncAP(%)↑|$\mathbf{FD_VGG}$↓|Resemblyzer↑|$\mathbf{CLAP_A}$↑|
> |------|---------------|-----------------|------------|-----------------|
> |MMAudio-S1|65.53|4.77|0.8185|0.5063|
> |MMAudio-S2|68.72|3.05|0.8360|0.5132|
> |MMAudio-S3|68.92|2.16|0.8568|0.5195|
> |FoleyGenEx (Ours)|69.38|**0.54**|0.9085|0.7216|
> |FoleyGenEx + AA (Ours)|**69.71**|**0.54**|**0.9128**|**0.7250**|
> |M-1|66.97|3.45|0.8430|0.5141|
> |M-2|68.90|2.91|0.8470|0.5224|
> |M-3|67.95|3.13|0.8677|0.7202|
>
> *Table 2. Comparison of Foley extension performance.$\mathbf{C_A}$:$\mathbf{CLAP_A}$.*
> |Conditions||||MultiFoley||MMAudio||FoleyGenEx||M-1||M-2||M-3||
> |----------|-|-|-|-------|-|------|-|---------|-|--|-|--|-|--|-|
> |$\mathbf{V_t}$|$\mathbf{Text}$|$\mathbf{A_r}$|$\mathbf{V_r}$|$\mathbf{C_A}$↑|$\mathbf{DeSync}$↓|$\mathbf{C_A}$↑|$\mathbf{DeSync}$↓|$\mathbf{C_A}$↑|$\mathbf{DeSync}$↓|$\mathbf{C_A}$↑|$\mathbf{DeSync}$↓|$\mathbf{C_A}$↑|$\mathbf{DeSync}$↓|$\mathbf{C_A}$↑|$\mathbf{DeSync}$↓|
> |$\checkmark$|$\checkmark$|||55.4|0.79|56.0|0.40|59.7|0.39|57.5|0.40|57.5|0.40|58.9|0.39|
> |$\checkmark$||$\checkmark$||59.6|0.78|57.0|0.46|61.8|0.40|58.1|0.46|58.2|0.44|60.9|0.42|
> |$\checkmark$||$\checkmark$|$\checkmark$|59.8|0.77|59.0|0.39|69.3|0.37|65.9|0.39|66.2|0.38|68.7|0.38|
> |$\checkmark$|$\checkmark$|$\checkmark$|$\checkmark$|64.3|0.77|60.7|0.38|71.2|0.36|70.0|0.37|70.2|0.37|70.9|0.36|

---

> ### Author Response · Authors · 2025-11-20
> **Part (3/5): Limited Performance Gains in Tables 2–3 (original paper): Root Cause & Evidence from Table 6 and Section 5.5 on Adverb Augmentation Performance**
>
> The modest performance improvements in Tables 2–3 stem from the **scarcity of adverb-containing samples** in the test sets:
> - VGGSound test set (Table 2): No test samples contain adverbs.
> - AudioCaps test set (Table 3): Only 12.6% (122/964) of samples include adverbs
>
> Thus, observed gains primarily come from training set expansion via adverb augmentation.
>
> For direct validation of adverb data’s impact:
> 1. **Table 6 (original paper)**: Evaluated MMAudio and FoleyGenEx on the 122 adverb-containing AudioCaps test samples—showcased improvements in distribution matching and semantic relevance.
> 2. **Section 5.5 (Subjective Evaluation)**: On 500 adverb-specific test samples, MMAudio trained with adverb augmentation outperformed the baseline in 386 cases, demonstrating stronger adverb-driven fine-grained control.

---

> ### Author Response · Authors · 2025-11-20
> **Part (4/5): Subjective Evaluation of Adverb-Audio Semantic Relevance**
>
> - Addressed via a "good-same-bad" human study (Section 5.5) using a specially designed adverb test set.
> - The evaluation directly verifies whether the **semantic of adverbs in text** is perceptually reflected in generated audio—complementing automated metrics with human perceptual analysis.

---

> ### Author Response · Authors · 2025-11-20
> **Part (5/5): Hyperparameter Design & Rationale**
>
> All hyperparameters are collaboratively optimized for multimodal audio filling, training-inference consistency, and performance:
>
> |Hyperparameter|Setting|Rationale|
> |--------------|-------|---------|
> |Audio latent masking ratio|70%-100%|Ensures sufficient audio filling learning signals, enables dynamic random-length masking to realize variable reference audio length, and avoids over-reliance on unmasked features.|
> |Video/text condition discard rate (CFG training)|10%|Balances multimodal guidance and generalization. This setting aligns with MMAudio.|
> |Reference audio discard rate|30%|Acts as audio-specific CFG, set higher than the 10% video/text rate to introduce reference audio guidance while preserving baseline performance.|

---

### Author Response · Authors · 2025-12-04
**Summary for the Area Chair**

We sincerely thank all four reviewers for their time, thoughtful comments, and constructive feedback. We have carefully addressed each concern during the rebuttal, and the manuscript has been substantially improved as a result:

### **Response to Reviewer 714K**
The reviewer raised questions about innovation, performance gains, component isolation, subjective validation, and hyperparameters. We addressed these by:
- Clarifying that our conditional injection + multimodal masking enables 4 new tasks (not incremental tweaks), and adverb augmentation supports unique semantic nuance control.
- Explaining minor metric gains stemmed from low adverb coverage in test sets, and supplementing adverb-specific test set results + human evaluation.
- Providing ablation experiments for modules (masking/adverb augmentation) and detailing hyperparameter designs.

### **Response to Reviewer ifQt**
The reviewer inquired about technical contributions, implementation details, and follow-up questions on configurations/feature shifts. We addressed these by:
- Adding ablation experiments to isolate component effects (e.g., masking vs. injection).
- Detailing AC-VTA latent processing (training-inference design) and latent inversion editing workflows.
- Defining S1/S2/S3 (fake reference video configurations) and verifying the validity of inference-time latent summation via ablation.

### **Response to Reviewer nxbb**
The reviewer focused on architectural clarity, novelty, data augmentation, and audio quality. We addressed these by:
- Specifying MMAudio modifications (conditional injection/masking) and defining core features ($\mathbf{F_S}$, $\mathbf{F_{RS}}$).
- Contrasting our conditional injection with standard methods (e.g., FILM layers) via tables + ablation.
- Validating speed augmentation (moderate scaling, 97% usability) and supplementing spectral comparisons + sync latency reductions (>100ms).

### **Response to Reviewer KTNo**
The reviewer questioned adverb augmentation’s necessity, adverb selection, and architectural modifications. We addressed these by:
- Explaining weak general-set gains (low adverb samples) and supplementing 500-case human evaluation + real-scenario examples.
- Detailing adverb selection (GPT-4 extraction → frequency filtering → antonym pairing).
- Adding a dedicated section listing MMAudio improvements (conditional injection, masking, loss function).

We have fully addressed all reviewers’ concerns, and the revised manuscript is more rigorous, clear, and complete. We appreciate the constructive engagement and hope you will consider the strengthened submission.

---

### Meta-Review · Area_Chair_2k5C · 2026-01-07

**Summary:**

The reviewers agreed that controllable and well-synchronized video-to-audio generation is an important and practical problem, and they appreciated the effort to unify multiple control modes in a single framework. However, several concerns informed the decision. (1) Multiple reviewers questioned the level of technical novelty, viewing the method primarily as a well-engineered integration of existing approaches (e.g., MMAudio-style diffusion, conditioning, and data augmentation) rather than a clear technical contribution. (2) There were concerns about clarity. The architecture, training pipeline, and differences from prior work were initially hard to follow. (3) Reviewers were not fully convinced by the evidence for “semantic precision,” particularly the adverb-based augmentation. (4) A methodological concern was raised about the mismatch between training and inference in the AC-VTA setting.

**Reviewer Concerns:**

***What was improved by the rebuttal***

Added ablations to isolate masking strategies and the effect of adverb augmentation, plus an adverb-focused subjective study.

Provided clearer definitions of task settings and models, and more detail on conditional injection and editing workflows.

Addressed several clarity issues and added comparisons to support synchronization claims.

***Remaining concerns***

The technical novelty is limited. While the system is well-engineered and integrates multiple components effectively, it does not introduce fundamentally new insights.

The motivation and representativeness of adverb-focused augmentation remain debatable, particularly given the low prevalence of adverbs in standard benchmarks. Although the curated test set and user study are helpful, they may not fully convince skeptical reviewers.

The potential train–test mismatch introduced by inference-time summation (not used during training) remains a non-trivial methodological concern.

**Reviewer Scores:**

Reviewer 714K (score 6): Likely to stay around 6. The rebuttal provides additional ablations, but novelty concerns would likely keep the score from moving much higher.

Reviewer ifQt (score 4): While many questions were answered, the reviewer’s remaining concern about train–test mismatch appears substantive; the score would likely remain 4 (or at best move slightly upward if fully satisfied, but that seems uncertain).

Reviewer nxbb (score 4): Likely to remain 4. The rebuttal improves architectural explanations and provides validation for augmentation, but the reviewer’s skepticism about novelty and perceptible quality gains appears only partially addressed.

Reviewer KTNo (score 4): Likely to remain 4. The rebuttal adds motivation and references, but the reviewer explicitly remained unconvinced that the adverb-focused argument is empirically supported or representative.

While the rebuttal improves the submission, it is unlikely to decisively shift the overall reviewer consensus toward acceptance.

---

### Decision · Program_Chairs · 2026-01-26

Reject